# A novel method to test non-exclusive hypotheses applied to Arctic ice projections from dependent models

R. Olson[1,2,3], S.-I. An[1], Y. Fan[4], W. Chang [ID] [5], J.P. Evans [ID] [6] & J.-Y. Lee [ID] [2,7]

A major conundrum in climate science is how to account for dependence between climate models. This complicates interpretation of probabilistic projections derived from such models. Here we show that this problem can be addressed using a novel method to test multiple non-exclusive hypotheses, and to make predictions under such hypotheses. We apply the method to probabilistically estimate the level of global warming needed for a September ice-free Arctic, using an ensemble of historical and representative concentration pathway 8.5 emissions scenario climate model runs. We show that not accounting for model dependence can lead to biased projections. Incorporating more constraints on models may minimize the impact of neglecting model non-exclusivity. Most likely, September Arctic sea ice will effectively disappear at between approximately 2 and 2.5 K of global warming. Yet, limiting the warming to 1.5 K under the Paris agreement may not be sufficient to prevent the ice-free Arctic.

[1] Department of Atmospheric Sciences, Yonsei University, Seodaemun-gu, Seoul 03722, South Korea. [2] Center for Climate Physics, Institute for Basic Science, Tonghapgigyegwan Building, Busandaehak-ro 63 beon-gil 2, Geumjeong-gu, Busan 46241, South Korea. [3] Pusan National University, Geumjeong-gu, Busan 46241, South Korea. [4] School of Mathematics and Statistics, UNSW Australia, Room 2055, Red Center, Sydney 2052, Australia. [5] Division of Statistics and Data Science, Department of Mathematical Sciences, University of Cincinnati, 5516 French Hall, 2815 Commons Way, Cincinnati, OH 45221-0025, USA. [6] Climate Change Research Center and ARC Center for Excellence in Climate Extremes, UNSW Australia, 4th Level Mathews Building, Sydney 2052, Australia. [7] Research Center for Climate Sciences, Pusan National University, Room 1113, Tonghapgigyegwan Building, Busandaehak-ro 63 beon-gil 2, Geumjeong-gu, Busan 46241, South Korea. Correspondence and requests for materials should be addressed to S.-I.A. (email: sian@yonsei.ac.kr)

Complex mathematical models are a key tool for providing projections of future climate and environmental changes[1–4]. Yet, it is well known that such models are dependent, which complicates extracting probabilistic projection information from multi-model ensembles[5–13]. Model dependence can be defined rather qualitatively as sharing of code or belonging to the same modeling group[5,6]. More quantitative definitions involve correlation[5,7,8], the concept of conditional probability[9], and distance between models in physical[6,10] or some transformed low-dimensional space[11]. Ignoring the dependence is expected to disproportionately pull future probabilistic projections towards clusters of similar dependent models. In the worst-case scenario, the future projections may be centered on output from a large group of models which rely on the same erroneous code, yet little weight may be assigned to an independent correct model. This is one of the reasons Intergovernmental Panel on Climate Change (IPCC) has outright removed any probabilistic global multi-model climate projections from its Fifth Assessment Report[12].

Fortunately, several new studies break important new ground[5–11]. These results indeed show that ignoring model dependence can result in overconfidence and bias in the future projections[5,6]. Yet, these methods typically treat model dependence in a simplistic way, without considering higher-order dependence (e.g., dependence between three or more models) or relying on probability theory. Moreover, many studies additionally assume exclusivity, i.e., that only one of the models is correct, e.g., refs. [14–16]. Hence, "the concept of independence has been frequently mentioned in climate science research, but has rarely been defined and discussed in a theoretically robust and quantifiable manner"[9].

Here, we first mathematically define model dependence and model exclusivity. Then, we present a novel statistical approach that can test multiple dependent non-exclusive statistical hypotheses, and make probabilistic projections under such hypotheses using Markov chain Monte Carlo (MCMC). The new approach differs from the traditionally used Bayesian model averaging (BMA), which assumes that hypotheses are exclusive. We apply the new method to provide the first multi-model probabilistic projections of the global mean surface temperature (GMST) change from the preindustrial period at which September Arctic sea ice will effectively disappear (thereafter called GMST change to melt). The projections are based on the output of the Coupled Model Intercomparison Project phase 5 (CMIP5) models. The method considers both model dependence and skill at capturing known present-day and uncertain future sea ice metrics. GMST change to melt, along with many other statistical model parameters, is jointly estimated using MCMC. Before making actual projections, we calibrate the method in a suite of observation system simulation experiments to provide approximately correct coverage of the 90% posterior credible intervals.

## Results

**Defining relevant statistical concepts**. Given the confusion in the literature, we first set out to define independence and exclusivity of events through their joint probability. Assume there are $n$ statistical hypotheses, and $H_1, …, H_n$ are events corresponding to each hypothesis being true. These hypotheses are tested using some observations $\mathbf{y}$ or a random variable $\mathbf{Y}$ representing the state of the dynamical system. In what follows we use a short-hand notation $\mathbf{Y}$ to mean the event of observing the underlying random variable $\mathbf{Y}$ taking a value of $\mathbf{y}$. Joint probability of two hypotheses $H_i$ and $H_j$ given the observations is defined as

$$P\left(H_i \cap H_j | \mathbf{Y}\right). \qquad (1)$$

If these hypotheses are conditionally independent given $\mathbf{Y}$, the joint probability is the product of the conditional probabilities[9]:

$$P\left(H_i \cap H_j | \mathbf{Y}\right) = P(H_i | \mathbf{Y}) P\left(H_j | \mathbf{Y}\right). \qquad (2)$$

Similar result holds for the combination of $n$ multiple hypotheses $H_1, H_2, …, H_n$. They are independent given $\mathbf{Y}$ if[17]:

$$P\left(H_{i_1} \cap H_{i_2} \cap \cdots \cap H_{i_r} | \mathbf{Y}\right) = P\left(H_{i_1} | \mathbf{Y}\right) P\left(H_{i_2} | \mathbf{Y}\right) \cdots P\left(H_{i_r} | \mathbf{Y}\right) \qquad (3)$$

for every $r$-combination of the hypotheses, and $r = 2, 3, …, n$. The hypotheses are conditionally dependent if the expression above does not hold. In the general case where hypotheses can be conditionally dependent:

$$\begin{aligned} P(H_1 \cap H_2 \cap \cdots \cap H_n | \mathbf{Y}) \\ = P(H_1 | H_2 \cap \cdots \cap H_n \cap \mathbf{Y}) P(H_2 | H_3 \cap \cdots \cap H_n \cap \mathbf{Y}) \cdots \\ P(H_{n-1} | H_n \cap \mathbf{Y}) P(H_n | \mathbf{Y}). \end{aligned} \qquad (4)$$

If the hypotheses are exclusive, then all the joint probabilities are zero:

$$P\left(H_i \cap H_j | \mathbf{Y}\right) = 0 \quad \text{if} \quad i \neq j \qquad (5)$$

and similarly for higher-order combinations. There are some relationships between the terms "exclusive" and "dependent". For example, exclusive events are dependent. However, dependent events do not have to be exclusive.

These concepts can be applied to climate modeling. While any overall hypothesis about model correctness is "false (and, it could be argued, not even approximately true)"[18], hypotheses may be formulated for model adequacy for a particular purpose. An example hypothesis may be that a model is adequate for predicting GMST change by year 2050 relative to the preindustrial period under a particular emissions scenario within a given uncertainty. In case of hypotheses representing physical models being adequate for a particular purpose, both assumptions of independence and exclusivity are likely wrong. Consider, for example close-cousin climate models, such as IPSL-CM5A-LR and IPSL-CM5B-LR. Defining probability as degree of belief in a model being correct for a specific modeling purpose, a climate scientist may assign a higher conditional probability to IPSL-CM5A-LR if s/he were informed that its cousin IPSL-CM5B-LR is correct. This is inconsistent with independence (where a fact of one model being correct has no bearing on the conditional probability of any other model), and with exclusivity (where probability of IPSL-CM5A-LR would be zero if its cousin model was known to be correct).

**Law of total probability for non-exclusive events**. We now discuss BMA and extend it to properly accounting for model dependence. BMA is a popular procedure to make probabilistic projections[14–16,19–24]. BMA is based on the law of total probability applied to a continuous random prediction variable of interest $A$, several exclusive hypotheses (models of reality) $H_1, …, H_n \in S$, and a sample space $S$. Here, an $i$th hypothesis refers to the event $H_i$ that the $i$th model is adequate for a particular modeling purpose. BMA states that

$$p(A | \mathbf{Y} \cap S) = p(A | \mathbf{Y}) = \sum_{i=1}^{n} p(A | H_i \cap \mathbf{Y}) P(H_i | \mathbf{Y}), \qquad (6)$$

where lower-case $p$ represents a probability density function (pdf), upper-case $P$ is a probability, and all densities and probabilities are implicitly conditioned on $S$. With a slight abuse of the notation, $\cap \mathbf{Y}$ refers to intersection with the event that a certain

value of $\mathbf{Y}$ has been observed. According to Eq. (6), the resulting final projection pdf is a weighted mean of pdfs given each model, with weights representing relative probabilities of each model. We define sample space as at least one hypothesis being correct, i.e., $S = \cup H_i$. One important implicit assumption for BMA is that the events of individual hypotheses being correct are mutually exclusive, i.e., $P\left(H_i \cap H_j | \mathbf{Y}\right) = 0 \quad \forall i \neq j$, and similarly for all higher-order model combinations. This is in fact a binding assumption and it will lead to too much weight assigned to similar predictions created by highly dependent (similar) models. As we present later, it is possible for several good models to jointly match observations, and to be simultaneously correct.

We show that when the hypotheses are non-exclusive, the BMA is modified as follows (non-exclusive law of total probability):

$$
\begin{aligned}
p(A|\mathbf{Y}) = \ & \sum_{i=1}^{n} p(A|H_i \cap \mathbf{Y}) P(H_i|\mathbf{Y}) \\
& - \sum_{1 \le i < j \le n} p\left(A|H_i \cap H_j \cap \mathbf{Y}\right) P\left(H_i \cap H_j|\mathbf{Y}\right) \\
& + \sum_{1 \le i < j < k \le n} p\left(A|H_i \cap H_j \cap H_k \cap \mathbf{Y}\right) P\left(H_i \cap H_j \cap H_k|\mathbf{Y}\right) - \cdots \\
& + (-1)^{n-1} p(A|H_1 \cap H_2 \cap \cdots \cap H_n \cap \mathbf{Y}) P(H_1 \cap H_2 \cap \cdots \cap H_n|\mathbf{Y}).
\end{aligned}
\tag{7}
$$

We prove this formula using the so-called inclusion-exclusion principle in Supplementary Note 1. In the modified BMA the probability densities of future metric A given all model pairs are subtracted, weighted by the respective pair probabilities; conditional weighted densities given model triplets are added back again, and so forth. As Eq. (4) shows, the full law considers potential model dependencies up to order $n$. We propose to adopt a new term "model non-exclusivity", which refers to the fact that more than one model can be jointly consistent with reality. Non-exclusive model probabilities incorporate both models' skill at capturing observations, as well as inter-model dependencies (Eq. (4)).

**Probabilistic projections using the marginalization theorem.** Evaluating the terms in the full BMA equation (Eq. (7)) is non-trivial. However, if our hypotheses can be defined by some parameters $\boldsymbol{\theta}$ belonging to regions in parameter space, the problem simplifies. Specifically, we can use the marginalization theorem to obtain

$$
p(A|\mathbf{Y}) = \int p(A, \boldsymbol{\theta}|\mathbf{Y}) d\boldsymbol{\theta}.
\tag{8}
$$

The key idea is that if sample space is appropriately defined, this approach is equivalent to adding up all models interactions in Eq. (7). We stress that this also considers all model interactions of up to order $n$ (Eq. (4)). As we will show later, this approach is tractable, and can be implemented with relative ease using MCMC[25,26]. MCMC is a method to sample from a joint distribution of parameters using Markov chains. The chain of the prediction variable A provides a way to estimate its probability density. The fraction of samples within an intersection of regions associated with any hypothesis combination naturally provides a tractable way to calculate the probability for that combination.

**Application to projections of Arctic sea ice.** We make CMIP5 multi-model projections of GMST change to melt, using present-day September Arctic sea ice extent (SIE), as well as the recent trend in SIE with respect to GMST (SIE sensitivity), as constraints. For the ice-free Arctic, we use the SIE cut-off of 1 million km$^2$ to be consistent with previous work[27–30]. Several studies address future sea ice projections[10,27–29,31,32], but few in a

probabilistic way[30,33–36]. Our work can be seen as a first attempt to provide probabilistic projections of GMST change to melt, while explicitly accounting for all model dependencies. Our data consist of autocorrelated annual time-series of present-day (1979–2017) September SIE from 31 CMIP5[37] climate models $\mathbf{y}_i$ with unknown means $\mu_i$ ($i = 1, 2, \ldots, n$ is model index), corresponding observations $\mathbf{y}_o$ with an unknown random mean $\mu_o$, and the deterministic future GMST change to melt for the representative concentrations pathway (RCP8.5) scenario from each model $z_i^*$. In addition, we use the sensitivity of SIE to GMST (e.g., trend from linear regression) from each model $u_i$ and corresponding observations $u_o$. The list of models and their institutions is shown in Supplementary Table 1. Let $\mathbf{y} = (\mathbf{y}_o, \mathbf{y}_1, \mathbf{y}_2, \ldots, \mathbf{y}_n)$ be the collection of observed and model SIE means, $\mathbf{Y}$ be the event that the underlying random variable $\mathbf{Y}$ takes the value of $\mathbf{y}$, and $\boldsymbol{\mu} = (\mu_o, \mu_1, \ldots, \mu_n)$ be the collection of true observed and model present-day means. We assume that each model and observed SIE time series is modeled by a sum of a linear term and an autoregressive model of order 1 [AR(1)], with the only uncertain parameters being the means of the linear terms (see Methods). Thus, we assume that the slopes and the AR(1) parameters are fixed at estimated values, in the interest of reducing computational cost and complexity. Our projection random variable of interest $z^*$ is the future GMST change to melt.

The event for each model adequacy hypothesis can be defined in terms of the random parameters belonging to a region in $\mathbb{R}^m$ where $m$ is the number of parameters

$$
H_i \equiv \mu_i \in \left[\mu_o \pm \Delta_\mu\right] \cap z_i^* \in \left[z^* \pm \Delta_z^*\right] \cap u_i \in \left[u_o \pm \Delta_u\right],
\tag{9}
$$

where $\Delta_\mu$, $\Delta_z^*$, and $\Delta_u$ are uncertain random tolerance ranges for the hypotheses. In other words, hypothesis $H_i$ states that the present-day SIE mean for model $i$ is within some distance from the observed mean, its temperature sensitivity is within some distance from the observed sensitivity, and future GMST change to melt is, too, within some distance of the actual unknown GMST change to melt. Originally, we were going to use fixed tolerances in our analysis, which seemed as a more natural choice in the beginning. However, during validation of the method, this resulted in flat posterior pdfs with sharp cutoffs for a projection variable of interest. This lead us to make the tolerances uncertain parameters. Specifically, we assume that the tolerances follow half-normal (or truncated normal) distributions

$$
\Delta_\mu \sim N^+\left(0, \left[f \sigma_\mu\right]^2\right),
\tag{10}
$$

where $f$ is deterministic scalar error expansion factor, which can be calibrated using cross-validation, and $\sigma_\mu$ is deterministic sample standard deviation of differences between each model's present-day mean and next-closest model's mean. This standard deviation is the same for all models. We use this quantity in the tolerance parametrization since it can be thought of as a measure of model error. Thus, broadly speaking a model is correct if it is within some model-error-informed tolerance away from the observations. We use the half-normal distribution because the tolerances cannot be negative, and because it results in reasonably looking posterior pdfs. Testing other prior distributions for the tolerances is left to future work. The $f$ factor is introduced in order to correct for potential overconfidence. Here, $f \sigma_\mu = 0.50$. The distributions for $\Delta_z^*$ and $\Delta_u$ are defined similarly in terms of $\sigma_z^*$ and $\sigma_u$, which are defined analogously to $\sigma_\mu$ but for the future GMST change to melt and the sea ice sensitivity, respectively. Here, $f \sigma_z^* = 0.41$ and $f \sigma_u = 0.47$. The events for hypothesis combinations are simply defined by intersections of the regions for constituent hypotheses. We use the mean SIE and its sensitivity to constrain the models because we find considerable

relationships between these variables and the GMST change to melt (Supplementary Fig. 1). Moreover, previous research[28] shows that in climate models the present-day mean SIE is more strongly correlated to the Arctic ice-free year under the RCP8.5 scenario than other variables, such as mean annual sea ice volume, September sea ice trend, mean seasonal cycle of SIE, or the thin ice SIE. As previously, we define our sample space as the union of all hypotheses: $S = \cup H_i$. The sample space can also be thought of as a region in the $m$-dimensional space.

The probabilities can be calculated using Eq. (8) (reformulated in different variables) and Bayes' theorem:

$$
\begin{aligned}
p(z^*|\mathbf{Y}) &= \int p(z^*, \boldsymbol{\mu}, \Delta_\mu, \Delta_z^*, \Delta_u | \mathbf{Y}) \mathrm{d}\boldsymbol{\mu}\,\mathrm{d}\Delta_\mu \mathrm{d}\Delta_z^* \mathrm{d}\Delta_u \\
&\propto \int p(\mathbf{Y}|z^*, \boldsymbol{\mu}, \Delta_\mu, \Delta_z^*, \Delta_u) p(z^*, \boldsymbol{\mu}, \Delta_\mu, \Delta_z^*, \Delta_u) \mathrm{d}\boldsymbol{\mu}\,\mathrm{d}\Delta_\mu \mathrm{d}\Delta_z^* \mathrm{d}\Delta_u \\
&= \int p(\mathbf{Y}|\boldsymbol{\mu}) p(z^*, \boldsymbol{\mu}, \Delta_\mu, \Delta_z^*, \Delta_u) \mathrm{d}\boldsymbol{\mu}\,\mathrm{d}\Delta_\mu \mathrm{d}\Delta_z^* \mathrm{d}\Delta_u \\
&= \int p(\mathbf{Y}|\boldsymbol{\mu}) p(\Delta_z^*) p(\Delta_\mu) p(\Delta_u) p(z^*, \boldsymbol{\mu}) \mathrm{d}\boldsymbol{\mu}\,\mathrm{d}\Delta_\mu \mathrm{d}\Delta_z^* \mathrm{d}\Delta_u \\
&\propto \int p(\mathbf{Y}|\boldsymbol{\mu}) p(\Delta_z^*) p(\Delta_\mu) p(\Delta_u) 1_S \mathrm{d}\boldsymbol{\mu}\,\mathrm{d}\Delta_\mu \mathrm{d}\Delta_z^* \mathrm{d}\Delta_u .
\end{aligned}
$$

(11)

According to Bayes' theorem[38], the posterior density of parameters given the observations is proportional to the product of the likelihood of the observations given the parameters $p\big(\mathbf{Y}|z^*, \boldsymbol{\mu}, \Delta_\mu, \Delta_z^*, \Delta_u\big) = p(\mathbf{Y}|\boldsymbol{\mu}) = p(\mathbf{Y} = \mathbf{y}|\boldsymbol{\mu})$ (this likelihood, which depends only on $\boldsymbol{\mu}$, is defined in the Methods section), and the prior belief in the model parameters $p(z^*, \boldsymbol{\mu}, \Delta_\mu, \Delta_z^*, \Delta_u)$. Here we decompose the prior into priors for individual components assuming independence. Furthermore, $p(\Delta_\mu)$ follows Eq. (10), $p(\Delta_z^*)$ and $p(\Delta_u)$ are defined similarly, and the prior for other parameters $p(z^*, \boldsymbol{\mu}) \propto 1_S$ is an indicator function for membership in the set $S$. (The prior is uniform over set $S$, and 0 outside of the set.) We explore the joint pdf presented in the second term of Eq. (11) using MCMC. For each posterior MCMC chain draw $l$ of parameters $z^{*(l)}, \boldsymbol{\mu}^{(l)}, \Delta_\mu^{(l)}, \Delta_z^{*(l)}$, and $\Delta_u^{(l)}$ we compute a value of binary variable $h_i^{(l)}$:

$$
\begin{cases}
h_i^{(l)} = 1 & \text{if} \quad \mu_i^{(l)} \in \left[\mu_o^{(l)} \pm \Delta_\mu^{(l)}\right] \cap z_i^* \in \left[z^{*(l)} \pm \Delta_z^{*(l)}\right] \cap u_i \in \left[u_o \pm \Delta_u^{(l)}\right] \\
h_i^{(l)} = 0 & \text{otherwise},
\end{cases}
$$

(12)

which is an indicator of whether the $i$th hypothesis is true or not. The joint hypotheses are correct if all constituent hypotheses are correct, e.g.,

$$
h_{ijk}^{(l)} = h_i^{(l)} \wedge h_j^{(l)} \wedge h_k^{(l)} .
$$

(13)

Model probabilities for any combination (e.g., $p(H_i)$ or $p(H_{ijk})$) are obtained from the MCMC chain using relative frequency of samples falling into each hypothesis region (or hypothesis combination region).

We first test the method using one-at-a-time observation system simulation experiments (Methods section). Here, we pretend each of the models is correct and use its output as observations one-at-a-time. We then exclude the true model from the model set and calculate the pdf for $z^*$ (GMST change to melt), which we compare to actual projections from the true model (Supplementary Figs. 2 and 3). In these experiments we choose the deterministic error expansion factor $f = 3$ such that the method gives correct coverage of the 90% posterior credible intervals for $z^*$. We then constrain all available models with real observations to make actual projections of GMST change to melt. We compare the method to the standard BMA, which does not account for the non-exclusive terms. To do that, we simply perform weighted sampling from conditional GMST change pdfs given each model being correct $p(z^*|H_i)$ using our MCMC chain.

We perform five projection experiments. These experiments explore the sensitivity of the results to observational datasets, and to the natural fraction of the recent SIE decline. The range of natural fractions considered here is roughly consistent with prior model-based studies which place it between 5 and 50%[39–41]. HadISST_r51_40p.nat uses 1979–2017 HadISST data for the SIE, the 51st realization of the HadCRUT4 temperature for the calculation of sea ice sensitivity over years 1979–2017, and assumes that 40% of the 1979–2017 SIE decline was natural. HadISST_r51_anthro experiment is exactly the same except is assumes that all of the recent decline was anthropogenic. NSIDC_r47_20p.nat uses NSIDC SIE observations, 47th realization of temperature observations, and assumes a natural contribution to the ice decline of 20%. NSIDC_r91_40p.nat uses NSIDC observations, the 91st realization of temperature observations, and assumes the natural contribution to the decline of 40%. There are large uncertainties regarding the magnitude of the true ice sensitivity given the short length of the observational records, and regarding the cause of the recent SIE decline[31,35,42]. One way to sidestep the problem is to just use the mean SIE constraint for the shorter 1979–2004 period which is not as severely affected by the recent melt. This motivates the NSIDC_mean_only experiment. This experiment uses NSIDC observations for SIE for years 1979–2004, and no SIE sensitivity constraint (see Eq. (18) for the formula for the pdf of GMST change for this experiment). To make sure the expansion factor $f = 3$ is reasonable for this experiment, we perform observational system simulation experiments using the 1979–2004 calibration period, and no SIE sensitivity constraint. Supplementary Figures 4 and 5 provide validation results for these experiments, and confirm that in this case $f = 3$ provides reasonable coverage of the 90% posterior credible interval.

In what follows we focus on the HadISST_r51_anthro experiment to illustrate the capabilities of the new method. However, pdf properties of GMST change to melt are provided for all experiments, and averaged across the experiments.

**Model hypothesis probabilities**. We remind the reader that the models are defined to be adequate for modeling the relationship between GMST and SIE if their present SIE means fall within a distance $\Delta_\mu$ of observed SIE mean, the future GMST changes to melt fall within tolerance $\Delta_z^*$ of the true GMST change to melt, and the sea ice sensitivities to temperature are within $\Delta_u$ of the observed sensitivity. We do not know the true GMST change to melt, but the method provides its pdf. To calculate hypothesis probabilities we integrate out this uncertainty. The tolerances are also estimated as part of the method, and their uncertainties are also integrated out. We provide the pdf of the tolerances for the HadISST_r51_anthro experiment in Supplementary Fig. 6. The present-day SIE tolerance $\Delta_\mu$ has a mean of 0.561 million km$^2$ and its 90% posterior credible interval ranges from 0.121 to 1.14 million km$^2$. Similarly, for the GMST change to melt, $\Delta_z^*$ has a mean of 0.474 K, with the 90% posterior credible interval from 0.11 to 0.963 K. Finally, the tolerance for sea ice sensitivity $\Delta_u$ has a mean of 0.515 million km$^2$ K$^{-1}$ and the 90% posterior credible interval from 0.098 to 1.07 million km$^2$ K$^{-1}$.

The model weights for HadISST_r51_anthro experiment are presented in Fig. 1. As it can be seen in the figure, a few models attain considerable weights, while many models receive zero or near-zero weights. The model with the highest weight (model 11, GFDL-CM3), has the mean present-day SIE of 6.43 million km$^2$, similar to the observed value of 6.40 million km$^2$ used in this experiment; and the sea ice sensitivity of $-2.87$ million km$^2$ K$^{-1}$, similar to the observations ($-3.01$ million km$^2$ K$^{-1}$). The model with the second highest weight is HadGEM2-CC, while MRI-

CGCM3 has the third highest weight. We note that the weights are a diagnostic of the method, and they are not needed to obtain the pdf of the future projections. Hence, the step of finding weights can be theoretically skipped if only the projections are needed.

Besides individual models weights, our approach provides joint model weights for all model pairs (Fig. 2). These weights are less than individual model weights, since they represent intersections of regions for hypothesis pairs. Note that these weights account for both model skill and dependence, since $P(H_i \cap H_j|\mathbf{Y}) = P(H_i|H_j \cap \mathbf{Y})P(H_j|\mathbf{Y})$. Notably, for the HadISST_r51_anthro experiment (Fig. 2a) the pair of 2nd and 11th models (ACCESS1.3 and GFDL-CM3) gets a high weight. These models produce relatively similar mean present-day SIE (5.62 and 6.43 million km$^2$), future GMST change to melt (both 2.12 K), and present-day SIE sensitivity to temperature change ($-3.08$ and $-2.87$ million km$^2$K$^{-1}$). These models have a similar ocean component: it is MOM4.1 for ACCESS1.3 and MOM4-based for

GFDL-CM3. However, the sea ice models are not the same: CICE4.1 for ACCESS1.3 and GFDL Sea Ice Simulator (SIS) for GFDL-CM3[43–45]. In addition, out of the top five most dependent pairs, none of the pairs belong to the same modeling family[13]. Thus, our results indicate that models may exhibit similar behavior for reasons other than coming from the same modeling center. These other reasons include equifinality (different parameters and modeling structures resulting in the same output by chance), subtler sharing of ideas, using the same observational datasets for input parameter calibration, or a random realization of internal variability. However, our approach accounts for the observed and modeled uncertainty in mean present-day SIE due to internal variability.

A unique feature of our method is the capacity to obtain joint hypothesis probabilities for any desired hypothesis combination. Figure 3 illustrates this by showing joint weight of combinations of models 11, 18, and 20 with all other models. This triplet is chosen because of the inter-dependencies between these models (Fig. 2). Specifically, these models are included in the top five most dependent model pairs. A combination of (2, 11, 18, 20) gets the highest combined weight (note also that the pair 2 and 11 is the most likely pair in the ensemble), and a combination (29, 11, 18, 20) gets the second highest combined weight (the pair 29 and 11 forms the second most likely pair in the ensemble).

Finally, the probability of having exclusively one model being correct is higher than the probability of more than one model being correct. Specifically, about 1,200,000 of 1,900,000 MCMC samples correspond to just one correct model (Supplementary Fig. 7a) for the HadISST_r51_anthro. Nonetheless, the amount of samples corresponding to more than one model being correct is still considerable. For the progressively increasing number of models the joint model probability tends exponentially to zero (Supplementary Fig. 7a).

Comparing the HadISST_r51_anthro and other experiments with two constraints to the NSIDC_mean_only experiment (where only one constraint is used), it appears that the model non-exclusivity may be more important in the one-constraint case. There are less instances of high pair weights once the second constraint is introduced (Fig. 2), and the non-exclusive model probabilities appear lower (Supplementary Fig. 7). Moreover, in the NSIDC_mean_only experiment the number of MCMC samples with an exclusively one correct model is smaller than in the rest of the projection experiments. This suggests that using multiple constraints (at least in the context of introduced

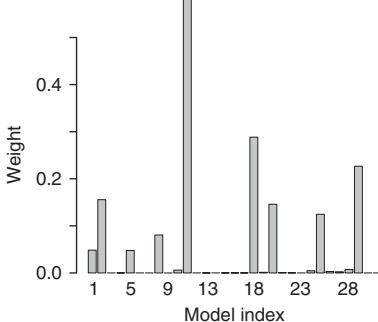

**Fig. 1** Individual Coupled Model Intercomparison Project phase 5 (CMIP5) climate model weights for the HadISST_r51_anthro experiment. The weights incorporate both model performance at capturing present-day mean September Arctic sea ice extent and its sensitivity to global mean surface temperature. The experiment assumes that recent sea ice extent decline was solely anthropogenic. This figure illustrates the capacity of the proposed method to find probabilities (weights) for any individual hypothesis in a given hypothesis set. In this case, each hypothesis is that regarding model adequacy at representing Arctic sea ice extent. Weighting is an optional component of the method; should one be interested only in prediction of a new variable given a set of hypotheses, weighting can be skipped entirely

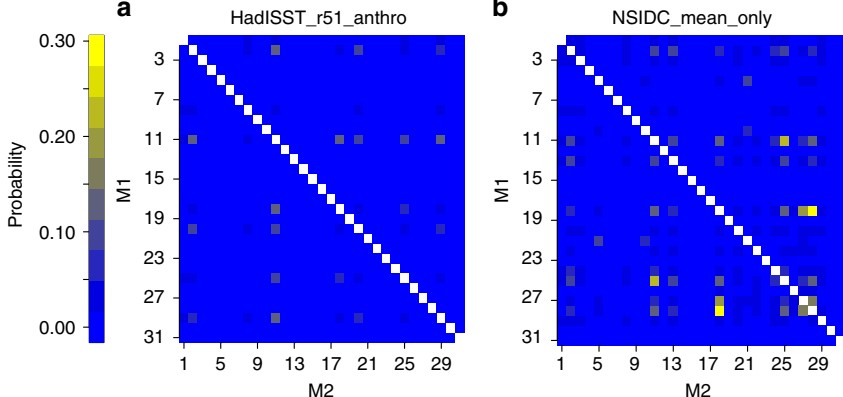

**Fig. 2** Joint model probabilities. **a** HadISST_r51_anthro and **b** NSIDC_mean_only experiments. NSIDC_mean_only experiment does not consider sea ice sensitivity to global mean surface temperature. y- and x axes index the first and the second climate model in the pair, while the joint probability is represented by color. The figure highlights the capability of the method to find joint probability of any pair of hypotheses in the set. There are less instances of high pair weights once the second constraint of sea ice sensitivity is introduced in the HadISST_r51_anthro experiment. This suggests that using multiple constraints may reduce the impact of hypothesis non-exclusivity

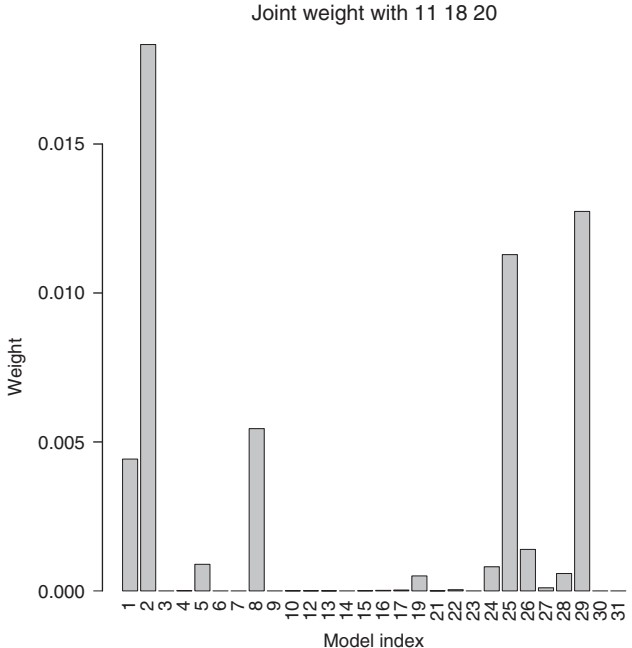

**Fig. 3** Combination model probabilities. Joint weights of each climate model in combination with models 11, 18, and 20. This figure illustrates the power of the method to find probability of any desired combination of hypotheses in a given set

methodology) may provide key to reducing the impact of non-exclusivity. If this is indeed the case, then performing standard BMA using a large number of constraints may give reasonable results even without accounting for non-exclusivity. Testing this hypothesis should be the subject of future research.

**GMST change at which Arctic sea ice will melt**. Our method provides pdfs for 36 parameters, however of main interest is the GMST change at which the Arctic ice will effectively melt in September. Figure 4a explores the effect of accounting for the non-exclusivity on the GMST to melt pdfs. It appears that the effect is relatively small for the case of two constraints (HadISST_r51_anthro with and without interactions). Yet, in the NSIDC_mean_only experiment, once the interactions are accounted for, the peak in GMST change to melt between 2 and 2.5 K decreases considerably, while the upper tail of the pdf becomes much fatter. Thus, the effects of accounting for interactions diminish in the case of using two constraints on the models. This is consistent with discussion in the previous section regarding introducing multiple constraints.

Figure 4b illustrates the differences between the pdfs under different assumptions about the observations, and about the fraction of the recent SIE decline caused by anthropogenic effects. For example, under the assumption of a fully-anthropogenic SIE decline, the HadISST sea ice dataset, and using the 51st realization of HadCRUT4 temperature observations, the pdf is relatively tight with the most likely GMST change to melt (mode) slightly above 2 K, and the 90% credible interval of (1.44, 3.17) K (Table 1). Keeping the HadISST SIE dataset, but assuming that 40% of the recent SIE decline was natural (HadISST_r51_40p.nat) changes the pdf considerably and results in the second peak above 4 K. This suggests that determining the cause of the recent SIE decline is fundamental in reducing the uncertainty about future GMST to melt projections. Yet, keeping the natural fraction of 40%, but switching to the NSIDC ice dataset and to the 91st realization of temperature observations substantially changes the

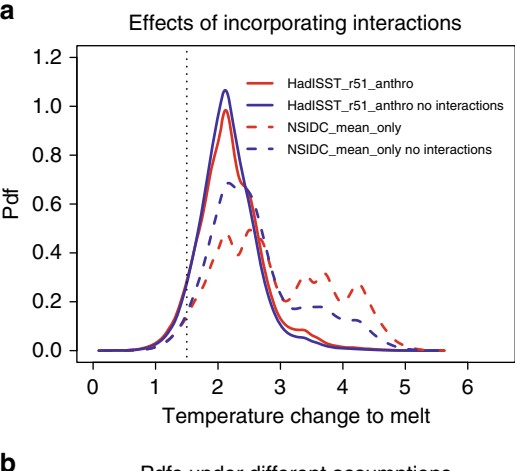

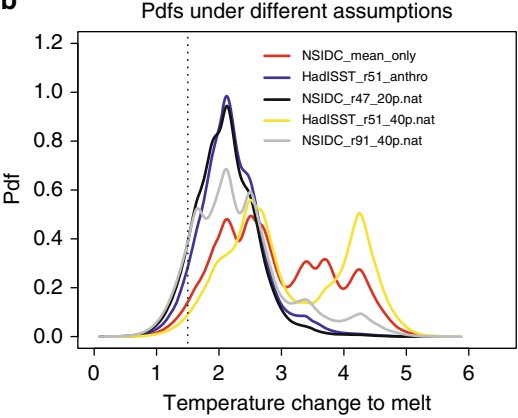

**Fig. 4** Probability density functions (pdfs) of global mean surface temperature change required for September Arctic sea ice to effectively vanish. **a** Pdfs for runs with and without interactions (to illustrate the effects of accounting for model interactions). **b** Pdfs from all runs accounting for interactions, to illustrate the impact of different assumptions. Vertical dotted line: lower desirable warming limit of 1.5° under the Paris agreement. The projections are sensitive to the datasets used, and to the assumptions about the cause of the recent Arctic sea ice decline. There is a distinct probability that keeping global warming below the 1.5° target of the Paris agreement may not be enough to stave off an essential disappearance of summer Arctic sea ice. The figure illustrates the capacity of the method to make predictions of a variable of interest conditioned on a set of non-exclusive hypotheses while accounting for all orders of hypothesis interactions

pdf once again (NSIDC_r91_40p.nat experiment). Specifically, the pdf is shifted to the left, and the second temperature peak between the 4 and 4.5 K is almost completely removed (Table 1). This illustrates the monumental impact the uncertainties in the observations can have on future Arctic sea ice projections. These effects have been pinpointed in previous work[34,35]. Finally, the pdf under the NSIDC sea ice observations, the 47th realization of temperature observations, and a natural fraction of the sea ice decline of 20% (NSIDC_r47_20p.nat), is again tight, with no second peak, a mean of 2.11 K, and the 90% posterior credible interval of (1.35, 2.92) (Table 1). Removing the sea ice sensitivity constraint results in a wide pdf with a relatively high mean (2.90 K). In most of the experiments, the lower bound of the 90% posterior credible interval is below 1.5 K. This indicates that there is a distinct probability of the Arctic losing essentially all of its summer ice even if we fulfill commitments to the lower 1.5 K warming limit of the Paris agreement. These results are in stark contrast to an almost zero probability of Arctic ice

**Table 1 Pdf properties of GMST warming to melt**

| Experiment | Mean (K) | Median (K) | Mode (K) | 90% credible interval (K) |
|---|---|---|---|---|
| NSIDC_mean_only | 2.90 | 2.73 | 2.51 | (1.60, 4.43) |
| HadISST_r51_anthro | 2.21 | 2.16 | 2.12 | (1.44, 3.17) |
| NSIDC_r47_20p.nat | 2.11 | 2.09 | 2.13 | (1.35, 2.92) |
| HadISST_r51_40p.nat | 3.12 | 2.87 | 2.51 | (1.73, 4.59) |
| NSIDC_r91_40p.nat | 2.35 | 2.22 | 2.12 | (1.32, 4.05) |
| Mean of all experiments | 2.54 | 2.41 | 2.28 | (1.49, 3.83) |

disappearance at 1.5 K warming suggested in previous work[33]. Yet, that work does not constrain the models by the sea ice sensitivity to temperature. On the other hand, our results broadly agree with a recent study that does use the sea ice sensitivity in correcting climate model biases[34]. Overall, the mean GMST change to melt for all experiments accounting for non-exclusivity is 2.54 K, with the 90% posterior credible interval from 1.49 to 3.83 K (Table 1).

Our results have been performed under the assumption of RCP8.5 emissions scenario. However, previous work has found a remarkably linear relationship between temperature and Arctic sea ice in various months in climate models/climate model ensembles[34,35,42]. Moreover, SIE metrics of individual models (at least for the annual mean), and entire CMIP5 ensemble (for September) have been found to not strongly depend on the driving emissions scenario[36,42, and others]. Thus suggests that current pdfs may be adequate not just for RCP8.5, but for a range of future emissions scenarios[35] where GMST monotonically increases with time. Validating our results using other emissions scenarios is subject of future work.

## Discussion

The non-mathematical understanding of our method is as follows. To provide pdf of the GMST change to melt, we randomly sample all GMST changes that fall within our sample space: GMST changes that are within some distance of at least one model whose present-day mean September SIE is within some distance of the observed mean, and whose present-day SIE sensitivity is also within some distance of the observed value. In addition, we also account for the uncertainty in present-day modeled and observed SIE, as well as for the uncertainty in distances themselves. We show that such an approach is equivalent to considering all model interactions of up to order $n$, where $n$ is the number of models.

Our results, specifically Eq. (7) indicate that BMA is limited in that it considers that models are exclusive representations of reality. This point has been previously discussed in the literature[22,46]. We show that using standard BMA can lead to biased projections in the case of a single observational constraint (Fig. 4a). In this case, the probability that more than one model is simultaneously correct is around 60% (Supplementary Fig. 7b). Yet, adding a second constraint substantially reduces this probability (Supplementary Fig. 7a). This illustrates that while the model non-exclusivity may be important, using multiple constraints may reduce its effect on model projections.

In the two-constraint cases, the most likely scenario is that of one and only one model (out of the 31 models in the ensemble) being adequate to represent reality. This raises a daunting question of usefulness of the model ensemble as a whole for making future projections. Should we not have restricted our sample space to a region with at least one model being adequate to represent relationship between sea ice and temperature, what would have been the probability of none of the models being adequate? To our knowledge, methods to quantify such a probability currently do not exist. Yet, determining the validity of the

ensemble as a whole is fundamental from the policy-making perspective. This should become focus of future research on this topic.

Our method can be compared with a recent study[11]. Specifically, that work uses singular value decomposition (SVD) and multidimensional scaling (MDS) to map present-day CMIP5 spatial model output, and corresponding observations into a 2D parameter space. Associated with each point in this space is a prediction property of interest (in this case climate sensitivity CS, as well as future regional temperature and precipitation changes), interpolated between model points. The study then proceeds to sample randomly from all points in the space within the convex hull of the models, with denser sampling close to observations, to construct a cumulative distribution function for CS, and pdfs for temperature and precipitation changes. The work claims their pdfs are just "resampled histograms of model behavior"[11]. However, we show using additional statistical analysis in Supplementary Note 2 (for their "Gaussian" experiment), that their approach is similar, and also considers all model interactions under some limiting statistical assumptions. Specifically, by using deterministic interpolation they assume a degenerate conditional probability for CS given a value of parameters in the 2D space. In addition, convex hull of the models is clearly a crude approximation to the probability space. Even if questions remain about the justification for their statistical model, this means that Bayesian multi-model probabilistic projections accounting for all model interactions ($2^{50} - 1 \approx 1.1 \times 10^{15}$ interactions) are already available to climate community. There are numerous differences between our work and that study. First, we consider dependence in both present-day and future model output. Second, we use time-series while that work uses spatial model output[11]. Third, we provide a statistical theory and method for finding non-exclusive hypothesis probabilities, and for prediction under such hypotheses that accounts for all hypothesis interactions.

Another relevant method is Bayesian model combination or ensemble BMA[22,47]. This method accounts for model non-exclusivity in a different way. Specifically, it considers a collection of augmented models $H_j^*$, where each augmented model represents a combination of original models $H_i$. It then performs standard BMA with the augmented models. However, such an approach suffers from the same problem as original BMA: it assumes exclusivity of model combinations. But if one model combination is correct, it does not preclude another model combination (e.g., a subset of the original combination) to be correct. We show here that the proper way to account for model interactions is to subtract model combination terms with an even number of models, and to add odd-number combinations following Eq. (7). By working with a parameter space (Eq. (8)) as opposed to model combinations (Eq. (7)), our approach can handle ensembles with a large number of combinations. This is because unlike the ensemble BMA we do not need to explicitly calculate the probability of each combination. Specifically, current work handles $2^{31}-1$ combinations, which is approximately 2.1 billion. Finally, previous ensemble BMA implementation does not

properly account for model dependence when evaluating model combination probabilities[22].

Our work is subject to important caveats. First, joint/combination model weights are dependent only on model and observed output in a low-dimensional space. As such, they do not explicitly consider model families[13], or sharing model code between institutions. While similar model output from unrelated models can result from subtle model dependencies such as sharing ideas or calibration datasets, it may also arise due to a random realization of internal variability[48]. Our code partially accounts for this by considering the uncertainty in the present-day model and observed SIE means. It has been previously shown, that when spatial information is considered, models from the same institution often produce similar output[11,13]. Hence, incorporating such information can be considered in future work. Dimension reduction methods used previously may constitute one useful avenue for action[11,49]. Second, we consider only two observational SIE datasets[50–52]. However, another popular dataset, Meier dataset, is based on the same satellite observations as NSIDC from year 1979[50,53]. Third, we do not consider the uncertainty in the autocorrelation, or standard deviation of the interannual present-day September SIE variability. While considering such uncertainty may present a substantial improvement, we refrain from doing this to drastically reduce the computational cost and complexity. Considering these uncertainties is subject of future work. Fourth, we use only a subset of available models and do not make use of multi-parameter model ensembles, or intermediate complexity Earth System models. Fifth, our tolerances for sea ice sensitivity are the same for all models, and do not account for the different internal variability in different models[54]. Ideally, this information should be incorporated in the sea ice sensitivity constraint. Note that this is not an issue with the mean SIE constraint, as there we sample from the pdfs of unknown population means. These pdfs take into account different internal variabilities of different models, e.g., if a particular model has a high internal variability this is expected to result in a broader pdf for the population mean of the SIE time series. Finally, our decision to formulate hypothesis tolerances using half-normal prior distributions is clearly subjective. We have chosen to make tolerances uncertain parameters because using fixed values resulted in flat pdfs with sharp cutoffs for a projection variable of interest in method validation experiments. We justify the prior distributions post-hoc by the fact that such priors lead to reasonable projection pdfs and model weights that gradually taper off further away from observations (Supplementary Figure 8). More rigorous determination of appropriate tolerance priors needs to be explored in future work.

In closing, we provide a statistically-robust Bayesian method to calculate probabilities of non-exclusive dependent hypotheses and their combinations; and to make predictions under such hypotheses. The approach accounts for $2^n - 1$ hypothesis combinations for $n$ hypotheses. We use this method to make projections of the GMST change from preindustrial (1861–1890) climatology at which the Arctic will lose almost all of its September ice using 31 non-exclusive climate models. Neglecting model non-exclusivity produces biased results in case of using just a single mean sea ice extent (SIE) constraint on the models, but the effects of non-exclusivity appear to diminish when a second constraint of SIE sensitivity to temperature is added. There is a distinct probability the sea ice may vanish below 1.5 K warming limit of the Paris agreement, even if 40% of the recent sea ice decline has been naturally-caused. The projections of GMST change to melt are sensitive to the assumptions about the observational datasets, and to the natural fraction of the recent sea ice melt. The overall mean for the GMST change to melt is 2.54 K, and the 90% posterior credible interval is (1.49, 3.83) K.

The study raises important questions about the usefulness of model ensembles (hypothesis sets) for making future projections. While the model runs used here were performed under RCP8.5 emissions scenario, the conclusions may hold more generally for a range of future emissions scenarios[35] where GMST continuously increases with time. Finally, by making parallels between our work and a previous study[11], we show mathematically that probabilistic regional climate projections and climate sensitivity estimates accounting for all model interactions are already available to climate community.

## Methods

**Model output and observations.** We use 31 CMIP5 climate models, utilizing the historical and future RCP8.5 first run from each model (Supplementary Table 1). This output has been obtained from the ESGF LLNL portal[55]. We first base model selection on the availability of output, which originally results in 33 climate models. We use the first run from each climate model. We then discard GISS-E2-H model as it reaches ice-free conditions (defined by <1 million km² September SIE) before present under the RCP8.5 scenario, and CSIRO-Mk3.6.0 as it has the highest September SIE of all models during the historical period 1979–2004, making it inconsistent with observations. We interpolate all sea ice concentration (sic) model output to a common 1° × 1° latitude-longitude grid using nearest-neighbor interpolation. Interpolating modeled sic to a common grid has been previously performed in the literature[29]. We calculate September SIE for years 1979–2017, and 2006–2099 as the total area of all cells in the Northern Hemisphere with sic > 0.15[56]. To obtain GMST change to melt we use the global mean surface atmospheric temperature difference between the 5 years centered on the year when Arctic first becomes ice-free, and the 1861–1890 preindustrial climatology for each model. These years were chosen because several models do not have data for the complete period 1850–1861. Before global averaging, we first bilinearly interpolate temperature fields to a 2° × 2° grid.

We use several observational sources for the Arctic September SIE. First, they include HadISST1 observations[52] for years 1979–2017. HadISST1 observations are in the form of monthly sea ice concentrations. We obtain SIE from these observations using the same procedure as used for the CMIP5 models. Second, we use version 3 NSIDC September SIE observations spanning years 1979–2017[57]. Third, we utilize NSIDC Sea Ice Extent version 1[50,51,58] for years 1979–2004. The differences in September SIE between the successive versions are small[57,59]. For GMST observations (used to calculate the sea ice sensitivity) we use different randomly chosen realizations of HadCRUT4 dataset[60]. We use different realizations of the same dataset as opposed to using different products, because the difference between HadCRUT4 1979–2010 GMST trend, and the GISS and NCDC datasets, is small compared to HadCRUT4 trend's own uncertainty range[60].

**Statistical model and detailed likelihood function.** We assume each present-day model SIE output, as well as observations can be modeled as sum of a linear trend and an AR(1) process:

$$\begin{cases} \mathbf{y}_o = \mu_o + \hat{k}_o \Delta \mathbf{t} + \boldsymbol{\varepsilon}_o \\ \mathbf{y}_i = \mu_i + \hat{k}_i \Delta \mathbf{t} + \boldsymbol{\varepsilon}_i \quad \forall i = 1, \ldots, n, \end{cases} \quad (14)$$

where $\hat{k}_o$ and $\hat{k}_i$ are observed and modeled sample slopes respectively, $\Delta \mathbf{t} = (t_1 - t_0, t_2 - t_0, \ldots, t_p - t_0)$ is the vector of centered years with $t_0$ being the mid-period year (e.g., 1998 for the 1979–2017 data, etc.), while $\boldsymbol{\varepsilon}_o = (\epsilon_{o,1}, \epsilon_{o,2}, \ldots, \epsilon_{o,p})$ and $\boldsymbol{\varepsilon}_i = (\epsilon_{i,1}, \epsilon_{i,2}, \ldots, \epsilon_{i,p})$ are residual AR(1) processes (representing interannual variability) where $p$ is the number of yearly datapoints. The AR(1) processes are defined as

$$\begin{cases} \epsilon_{o,t} = \hat{\rho}_o \epsilon_{o,t-1} + w_{o,t} \\ \epsilon_{i,t} = \hat{\rho}_i \epsilon_{i,t-1} + w_{i,t} \quad \forall i = 1, \ldots, n, \end{cases} \quad (15)$$

with $\hat{\rho}_o$ and $\hat{\rho}_i$ being the observed and modeled sample autocorrelations, $w_{o,t} \sim N(0, \hat{\sigma}_o^2)$ being the observed random noise term with the sample innovation standard deviation $\hat{\sigma}_o$ and $w_{i,t} \sim N(0, \hat{\sigma}_i^2)$ being the modeled random noise with the sample innovation standard deviation $\hat{\sigma}_i$ for each model $i$. The slopes $\hat{k}_o$ and $\hat{k}_i$ are obtained by linear regression, and the variability properties $\hat{\sigma}_o$, $\hat{\sigma}_i$, $\hat{\rho}_o$, and $\hat{\rho}_i$ are found by maximum likelihood.

To obtain the likelihood function for the observations $\mathbf{y}$ we assume that the random variable $\mathbf{Y}$ describing the historical climate and model SIE output is independent between the models, and real climate, given the present-day modeled and observed SIE mean $\boldsymbol{\mu}$. This is justified as internal variability in climate models and observations is known to be random. Under such assumption the likelihood

function can be decomposed as

$$
\begin{aligned}
p(\mathbf{Y}|\boldsymbol{\mu}) &= p\big(\mathbf{y}_o, \mathbf{y}_1, \mathbf{y}_2, \dots, \mathbf{y}_n | \mu_o, \mu_1, \mu_2, \dots, \mu_n\big) \\
&= p(\mathbf{y}_o | \mu_o, \mu_1, \mu_2, \dots, \mu_n) \times \prod_{i=1}^{n} p\big(\mathbf{y}_i | \mu_o, \mu_1, \mu_2, \dots, \mu_n\big) \\
&= p(\mathbf{y}_o | \mu_o) \times \prod_{i=1}^{n} p\big(\mathbf{y}_i | \mu_i\big) \\
&= p\big(\mathbf{y}_o | \mu_o, \hat{k}_o, \hat{\sigma}_o, \hat{\rho}_o\big) \times \prod_{i=1}^{n} p\big(\mathbf{y}_i | \mu_i, \hat{k}_i, \hat{\sigma}_i, \hat{\rho}_i\big).
\end{aligned}
\tag{16}
$$

Here, individual likelihood terms are standard AR(1) process likelihood functions[61]. For example,

$$
\begin{aligned}
p\big(\mathbf{y}_o | \mu_o, \hat{k}_o, \hat{\sigma}_o, \hat{\rho}_o\big) &= (2\pi \hat{s}_o^2)^{-1/2} \exp\left(-\frac{\epsilon_{o,1}^2}{2\hat{s}_o^2}\right) \times (2\pi \hat{\sigma}_o^2)^{-(p-1)/2} \\
&\quad \times \exp\left(-\frac{1}{2\hat{\sigma}_o^2} \sum_{j=2}^{p} w_{o,j}^2\right),
\end{aligned}
\tag{17}
$$

where $\hat{s}_o^2 = \hat{\sigma}_o^2 / (1 - \hat{\rho}_o^2)$ is stationary process variance for observed variability, and $w_{o,t} = \epsilon_{o,t} - \hat{\rho}_o \epsilon_{o,t-1}$, $t>1$ represent whitened observed variability. The likelihoods for model outputs $p(\mathbf{y}_i | \mu_i, \hat{k}_i, \hat{\sigma}_i, \hat{\rho}_i)$ are defined similarly; we omit them for brevity.

Spectral analysis strongly suggests that AR(1) is a reasonable approximation for climate models used here (Supplementary Fig. 8). For the spectral analysis we use time series of September SIE variability of CMIP5 models over years 1880–2004 around a lowess trend[62] with a span = 0.8. We choose this value because it appears to remove multidecadal variability from the trends.

**Pdf for GMST to melt for the NSIDC_mean_only experiment**. In the NSIDC_mean_only experiment, we modify Eq. (11) to account for the fact that we are only using a single SIE mean constraint as follows:

$$
\begin{aligned}
p(z^*|\mathbf{Y}) &= \int p\big(z^*, \boldsymbol{\mu}, \Delta_\mu, \Delta_z^* | \mathbf{Y}\big) \mathrm{d}\boldsymbol{\mu} \mathrm{d}\Delta_\mu \mathrm{d}\Delta_z^* \\
&\propto \int p(\mathbf{Y}|\boldsymbol{\mu}) p\big(z^*, \boldsymbol{\mu}, \Delta_\mu, \Delta_z^*\big) \mathrm{d}\boldsymbol{\mu} \mathrm{d}\Delta_\mu \mathrm{d}\Delta_z^* \\
&\propto \int p(\mathbf{Y}|\boldsymbol{\mu}) p\big(\Delta_z^*\big) p(\Delta_\mu) 1_S \mathrm{d}\boldsymbol{\mu} \mathrm{d}\Delta_\mu \mathrm{d}\Delta_z^*.
\end{aligned}
\tag{18}
$$

All terms of Eq. (18) are the same as previously defined. However, the individual hypotheses $H_i$, and the sample space $S$ are redefined so they no longer use the sea ice sensitivity constraint.

**Details of the observation system simulation experiments**. During the observation system simulation experiments for the GMST change to melt we use each model's output as pseudo-observations one-at-a-time. In the first set of experiments we use both SIE mean and SIE sensitivity constraints for years 1979–2017. For each perfect model, we explore the joint pdf specified in the second term of Eq. (11) using MCMC. We use 200,000 MCMC samples, with a burn-in of 50,000. We use $f = 3$ for the observation system simulation experiments, as this results in reasonable empirical coverage of the 90% posterior credible intervals. For each excluded model we compare the posterior pdf for the GMST change to melt to the value from the true excluded model (Supplementary Figs. 2 and 3). The GMST change from the true model, and the mean projection exhibit a considerable positive correlation, highlighting the skill of the method at projecting this quantity. We then perform a similar experiment with just the mean SIE constraint for years 1979–2004. Supplementary Figures 4 and 5 show validation results for this case.

We test whether the chain length and burn-in settings in the observation system simulation experiments are reasonable. Using such MCMC length in an additional estimation experiment similar to NSIDC_r91_40p.nat, we find that the differences between marginal pdfs for the GMST change in the short and the standard run are small. In addition, chains for all parameters for the short run appear to be reasonably stationary and not unduly influenced by the initial random seed. This suggests that these MCMC settings are reasonable for calibration of the 90% posterior credible interval.

**Details of the future projection experiments**. We explore the joint pdf in the second term of Eq. (11) using MCMC. For the MCMC, we use 2,000,000 samples with a burn-in of 100,000 for all experiments. Following the cross-validation experiment, we use $f = 3.0$ for all experiments. We run a second run of the NSIDC_r91_40p.nat experiment with a different random seed, and the diagnostics from the two runs are reasonably similar.

## Data availability

CMIP5 model output is available online from the ESGF LLNL portal at https://esgf-node.llnl.gov/projects/esgf-llnl/. NSIDC Sea Ice Index version 3 is available online from the NSIDC website at https://nsidc.org/data/seaice_index. The version 1 data is available from the authors upon request. HadISST1 sea ice concentrations are available online from the Met Office Hadley Center website at https://www.metoffice.gov.uk/hadobs/

hadisst/. HadCRUT4 temperature observations are located at https://crudata.uea.ac.uk/cru/data/temperature/. All other relevant data are available from the authors upon request.

## Code availability

Code used to implement the method is available from the authors upon request.

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

## Acknowledgements

For their roles in producing, coordinating, and making available the CMIP5 model output, the authors acknowledge the climate modeling groups, the World Climate Research Programme's (WCRP) Working Group on Coupled Modeling (WGCM), and the Global Organization for Earth System Science Portals (GO-ESSP). R. Olson and S.-I. An were supported by the Basic Science Research Program through National Research Foundation of Korea (NRF-2017K1A3A7A03087790 and NRF-2018R1A5A1024958). R. Olson and J.-Y. Lee also acknowledge support from the Institute for Basic Science (project code IBS-R028-D1). Y. Fan is grateful for the support by the Australian Research Council Centre of Excellence for Mathematical and Statistical Frontiers (ACEMS). Jason Evans acknowledges support from the Australian Research Council Centre of Excellence for Climate Extremes (CE170100023).

## Author contributions

R.O. developed the method with the help of W.C. and Y.F., coded the method, and analyzed the results. S.-I.A., W.C., Y.F., J.P.E. and J.-Y.L. oversaw the research and contributed to analysis and writing the paper. J.P.E. provided the original idea for the research.
