## [Peer Review File · Nature Communications]

Reviewers' comments:

Reviewer #2 (Remarks to the Author):

After reading over this paper several times, I still do not know what to make of it. The fundamental problem is how to project the probability distribution of future climate variables based on multiple climate models. Over about the past 15 years, there has developed quite a rich literature on this topic, some of it by statisticians using such concepts as Bayesian hierarchical models, some of it by climate scientists using more intuitive concepts, though all of it is heavily statistical in flavor. Quite a bit of this literature makes the assumption that climate models from different modeling groups are (statistically) independent, and this has been criticized on the grounds that it ignores possible sources of dependence such as different models from the same center being treated as independent models, or sharing of modeling concepts or computer code between different centers. There is some literature that has tried to address these issues directly, e.g. papers by Bishop and Abramowitz from 2013 (Climatic Change) and 2015 (Journal of Climate). So far, IPCC has not used probabilistic weightings of climate models in its main model projections, though I am inclined to suggest that is more because of the absence of any universally accepted approach rather than the specific technical point about independence. Nevertheless, the latter is the main issue the paper purports to solve. As an application of the proposed method, the authors consider future projections of the year in which September Arctic sea ice will effectively disappear, in particular making the point that with the authors' proposed reweighting, the right tail of the distribution becomes significantly longer, i.e. higher probability that the sea ice disappearance will be delayed beyond 2090. This is moderately interesting as a conclusion in its own right, though I am not sure it is a strong enough conclusion to carry the paper.

Regarding the "big picture" issues about multi-model ensembles, I have two broad comments:

1. The main point the authors make about BMA (Bayesian Model Averaging) is that one should not proceed as if the models were disjoint (the authors use the word exclusive). In other words, it's possible for more than one of the models to be correct at the same time. My reaction to this was to ask myself whether (a) all the previous authors who used BMA (including me) were doing it wrong, (b) whether the present authors are doing it wrong, or (c) whether the difference in point of view results from subtly different specifications of the problem. On reflection, I think the answer is (c). I pulled out and reread the two most statistical papers the authors cite, references 15 and 17. [15] is a summary of BMA written by and for statisticians and refers to the most common application of this technology - the choice among statistical models in a context such as regression. In this context, the problem is usually formulated in a way that assumes only one of the considered models can be correct. There's a technical issue here, in that it's quite common to apply BMA in contexts where one model may be nested inside another (i.e. the former derived from the latter by setting certain parameters to 0), but in Bayesian analysis with continuous priors, the posterior probability that a particular parameter is exactly 0 will always be 0, so the models are effectively disjoint. In this context I don't think there is any argument with the traditional interpretation of BMA. However, paper [17] takes a different approach - in the context of weather forecasting rather than long-term climate, but I don't think that affects the statistical issue under discussion - they do assume there are different physical forecasting models and one of them is "best" though without ever defining precisely (I don't think) what best actually means. So this looks like exactly the kind of BMA that the present authors are criticizing. But I think it comes down to how you define the problem. The present authors define a climate model as fitting the data if it satisfies a tolerance condition. I have issues with how that condition is defined, but I'll come to that later - my basic point is that, the way the authors set up the problem, it obviously does NOT exclude the possibility that more than one model may be correct, so a BMA that assumes the contrary would obviously be wrong. That seems to me the fundamental issue, which the authors may or may not have clear in their own minds, but they have certainly not articulated it clearly in the paper.

2. I also feel the authors have not clearly separated the issues of model independence and model disjointness (or exclusivity). Granted, nowhere in the paper did the authors directly confuse these two concepts (which I would describe as an elementary beginner's error of the sort I encounter

when teaching freshmen) but I still feel that they have confused them in some big-picture sense. To be specific: the first two paragraphs on page 4 are clearly about model dependence, but in the third, the emphasis switches without explanation to model exclusivity. Granted, the solution the authors are proposing would be simpler if the models were independent, but that doesn't seem to me the main point of the paper.

I have a whole host of minor issues about the paper, detailed below, but these seem to me the two major issues. I do believe the paper is making interesting points and could be rewritten in a way that would make it suitable for publication, but as things stand, I don't think it's making a clear-cut argument.

Here are the rest of my points about the paper:

3. Section 3.2 is about something called the inclusion-exclusion principle, but the authors never call it that. I can't really believe the authors have never heard that expression, but there's a wikipedia article about it if you care to look it up. So why this rather long-winded statement and even more long-winded proof (in the Supplementary Information section)? (To be explicit, the IE-principle is applied on the probability space of events that are intersected with the event A and conditioned on the random variable Y, but on that space, it's just the basic IE formula.)

4. However, as far as I can tell, formula (6) is never actually applied as given. Instead the authors immediately segue to equation (7), which is a much more familiar (and natural) formulation of the posterior distribution of A. So I'm still unclear about the purpose of the formula (6).

5. Section 3.4. Here my confusion was less about big-picture issues than the specifics of how the model is defined. What exactly is μ_i ? It seems it could be a parameter of dimension m for any m (line 154) and equation (8) makes else if the plus-minus operations are defined coordinatewise, but it's still not clear to me what physical quantity the parameter actually represents. Specifically, is it the 26-dimensional vector of annual means, or is it just a scalar - as much of the subsequent discussion, including figure 1, seems to suggest? It's quite common in this field of research to summarize a multi-year vector in terms of a single overall mean (even though the annual mean may be varying) and that would be by far the simplest interpretation of what the authors are doing, that also avoids the issue of whether sub-decadal effects such as El Ninos could bias the results. (That's why much of the climate literature uses 30-year means.) However it becomes clear later, this isn't the intended interpretation.

6. I also have a concern about the same notation (μ and μ^*) being used to describe physically different quantities - in one case, the proportion of sea ice, in the other, the year in which sea ice reaches a certain limit. It seems unnatural to me that one would use essentially the same statistical formulation to describe such different quantities. Maybe the authors think it is OK, but they haven't said much to justify it.

7. Page 10, equation (9). Now I'm getting REALLY confused. Why does this equation makes sense at all? The authors say they used the same idea in ref. [23] - well, I downloaded and tried to read that reference, but I couldn't see any equation that looked like (9). It would make sense to me if the authors simply defined a Delta that represented some qualitative notion of model equivalence (I'm not sure what that would be, but based on Figure 1 it looks as though a Delta of something like 0.5 on the scale of the x-axis would be reasonable) but why give it a half-normal prior distribution? And why define sigma as the "standard deviation of differences between each model's present-day mean and next-closest model's mean"? In other words, to define sigma for a single model you first have to look at ALL the other models (to determine which one is closest) and then define a standard deviation based on model differences? Please! Let's at least have a definition that allows me to determine, for ONE model and ONE set of observations, whether they are a good enough fit or not!

8. Equation (10). Here I have an issue about the assumed form of prior distribution for the μ 's and Delta's and, especially, whether this is consistent with the notion of model dependence, but some of this is discussed subsequently. See also the top of page 11.

9. Line 201: given the discussion at the beginning of this review, isn't the discussion of "standard BMA" just a straw-man comparison? In other words, standard BMA assuming disjointness may be justified when the hypotheses really are disjoint, but when they are not, isn't this just wrong?

10. Line 219 - this seems to be the first time in the whole paper that the acronym AMOC is used, so the authors should define it. (Atlantic Meridional Overturning Circulation, as I remembered when I looked at ref. 23.)

11. I don't understand lines 223-232 at all. Just rescaling the prior density doesn't change the posterior, because the posterior density always has to integrate to 1. So why do this? The authors themselves say "this weighting approach is incorrect..."

12. Lines 246-251. I'm not sure that the authors are justified in assuming that every instance in which two models produce similar results has a causal explanation ("subtler sharing of ideas", etc.). They could just be random chance. However, there might be a possible statistical test whether the number of near-coincidences in the model projections is greater than could be explained assuming model independence.

13. Section 3.6. The results are interesting but hardly earth-shattering. The key point, obviously, is how one interprets the anomalous projections of the model FIO-ESM. I would actually have appreciated more discussion of the physical reasons for this, e.g. whether the explanation offered in lines 215-216 is something that physicists consider important. As it stands, compared with what the authors call "standard BMA" (though I'm not sure the word standard is appropriate, for reasons explained earlier in this review) the 95th percentile of the distribution is postponed 8 years. But the lower tail is hardly changed at all, and for people worried about the pace of climate change, that is surely the more important end of the distribution.

14. Line 326: this is the first explicit acknowledgement that the authors are using a time series approach. They really need to carefully define all their assumptions at the beginning of the paper, not near the end.

15. Line 363: again, the first time that trend, autocorrelation and standard deviation (of the individual data values) have been mentioned.

16. Line 373: do I understand from this that the decision to end in 2004 was made deliberately, to avoid the increase of natural circulation variability? If so, does that bias the results in any way?

17. Line 385 - "removes the spurious mid-21st century .. peak ..." - there's still a bimodality, but it's less pronounced under the new approach.

18. Equation (13). As far as I can tell, these equations are just wrong. Or if you think they are correct, at least say what the formulas actually are (i.e. give explicit algebraic expressions). If you have an AR(1) model with trend, then the four parameters of the model are the autocorrelation, the innovations SD, the mean and the trend. Any textbook on time series analysis will give you the joint density of the observations under this model. But it doesn't look anything like (13).

19. Pages 20-21: I would have appreciated more discussion of the rationale behind the length of MCMC runs, including the burn-in. The authors use a much longer MCMC for the future projections part of the experiment than the past observations. Maybe they reached this conclusion after sound statistical testing, but it's not clear to me. Also, the burn-ins are much shorter than I would have expected - modern practice of MCMC often uses burn-ins of a quarter or even half of the entire MCMC run. Did the authors test the sensitivity of the results to this assumption?

20. The Supplementary Information was submitted in a form that made it very difficult to read. The figures were submitted as seven separate postscript files but there is no labelling which is which, and several did not display correctly in the postscript viewer I use (GSView). Why can't you convert to pdf and submit the entire SI as a single pdf file, just like the main paper?

Reviewer #3 (Remarks to the Author):

Review for the paper

Testing and Prediction Under Non-Exclusive Hypotheses: Application to Arctic Sea Ice Projections

In this article, the authors present a novel approach to account for model dependencies in multi-model ensemble analyses. This topic is relevant for the community as it has been shown that ignoring model dependence might lead to an overestimation of some results while others might be ignored even though they are more correct. The authors develop a new method based on the Bayesian model averaging.

The focus of this review is not on the statistical correctness or novelty of the approach presented here but on the results concerning the sea ice projections. Thus, in the following it is assumed that the statistical method is robust and the following points refer mainly to the sections 3.4-3.6.

1. My main concern is the fact that you look at the year when the Arctic will become ice-free. In my point of view it is more interesting to get to know the level of warming for an ice-free Arctic and not the year as this is dependent from the underlying emission scenario. The driving force for sea-ice evolution is given by temperature (or the amount of global warming) and thus by the emissions. It should be possible to use this approach in a similar way to calculate the pdf for global warming (and/or for CO2 emission)!? This could give much more information than your approach now.
2. How sensitive are your results on the choice of the time period for present-day? For me it is not intuitive to stop at 2004 (even though the historical simulations end in 2005). (Maybe present-day is just a wrong formulation.) How sensitive are the results to your choice of observational datasets? It has been shown that the observations differ largely (see [32]).
3. The joint hypothesis is defined as correct if all constituent hypotheses are correct (II.187-...) - would it be possible to define it in a different way? It could be that two h_i itself are wrong but together they are correct. Maybe a sentence about a possible expansion of this assumption could be added.
4. Supplementary Figure 1: Could you please explain the local minima in some of the pdfs. For me it is not clear why the pdfs look like this – further investigations could be made when forming groups of models with similar behavior and discuss the shape of the pdf. Do you have an idea of the underlying reasons for differing that largely? What is the difference of the underlying “true” models?
5. In paragraph 3.5, you discuss about the behavior of the model FIO-ESM because this is simulating present-day SIE well but loses ice much later than the other models. Could you say something about the sea ice thickness? As I understood you only look at the scenario RCP8.5, thus I would delete the sentence about temperature changes under RCP2.6 as this does not give an explanation. The explanation is given in the next sentence (II.219-222).
6. What do you mean with “model convergence” (I.229)?
7. II. 362-367 Could you give a suggestion on how to include these uncertainties and by how much the computational costs would increase (I don't want to see any numbers, but I have no feeling on how large the increase factor would be and how complicated an expansion would be...).
8. II.373-374 How would that change the result? It could reduce the observational uncertainty to use longer time-series (until 2017, instead of 2004 only) , but would include the trend. (See comment 2.)

Minor comments:

II. 20-21 a reference is missing for "postpone" - if you say the new year it would be nice to get the information on how many years the mean or the 95th percentile of the pdf is shifted

I.58 statistical meaningful is a very unprecise formulation- could you justify this somehow?

I.149 delete "the"

II.155-... could you specify the "uncertain random tolerance ranges"; what are your numbers for f and σ and Δ ? - you could maybe include a table in the supplemental material.

Supplementary Figure 2 the one-to-one line would be helpful to easily see the relationship.

I.240 "present-day" instead of "present day"

Fig.3 What do the colors mean in Fig.3?

I.262-264: How do I see that in Supplementary Fig.5?

I.336 "it" instead of "is"

I.390 write "[11]" instead of "Sanderson et al. (2015)"

Response to Reviewer #2

We thank the reviewer for their thorough assessment of the merits of the paper, and for very useful feedback. In fact, this is one of the most insightful reviews I (RO) personally have seen in many years working in the field. We address the reviewer's concerns by additional discussion (both in this response and in the main manuscript), and by an additional analysis with an extended observational period. We believe the new discussion can help the reviewer become more deeply familiar with the philosophy behind this paper, and where to place it in the context of prior work. In addition, we re-ran all the main experiments used in the original manuscript to eliminate potential code bugs. There are no changes except that the exclusive BMA 95% percentile for Arctic ice-free year goes up to year 2085 (instead of 2084). Our responses to the reviewer are marked by bold blue in this file, and using track changes in the main manuscript. All line references are for the manuscript version with tracked changes.

After reading over this paper several times, I still do not know what to make of it. The fundamental problem is how to project the probability distribution of future climate variables based on multiple climate models. Over about the past 15 years, there has developed quite a rich literature on this topic, some of it by statisticians using such concepts as Bayesian hierarchical models, some of it by climate scientists using more intuitive concepts, though all of it is heavily statistical in flavor. Quite a bit of this literature makes the assumption that climate models from different modeling groups are (statistically) independent, and this has been criticized on the grounds that it ignores possible sources of dependence such as different models from the same center being treated as independent models, or sharing of modeling concepts or computer code between different centers. There is some literature that has tried to address these issues directly, e.g. papers by Bishop and Abramowitz from 2013 (Climatic Change) and 2015 (Journal of Climate). So far, IPCC has not used probabilistic weightings of climate models in its main model projections, though I am inclined to suggest that is more because of the absence of any universally accepted approach rather than the specific technical point about independence. Nevertheless, the latter is the main issue the paper purports to solve. As an application of the proposed method, the authors consider future projections of the year in which September Arctic sea ice will effectively disappear, in particular making the point that with the authors' proposed reweighting, the right tail of the distribution becomes significantly longer, i.e. higher probability that the sea ice disappearance will be delayed beyond 2090. This is moderately interesting as a conclusion in its own right, though I am not sure it is a strong enough conclusion to carry the paper.

Regarding the "big picture" issues about multi-model ensembles, I have two broad comments:

1. The main point the authors make about BMA (Bayesian Model Averaging) is that one should not proceed as if the models were disjoint (the authors use the word exclusive). In other words, it's possible for more than one of the models to be correct at the same time. My reaction to this was to ask myself whether (a) all the previous authors who used BMA (including me) were doing it wrong, (b) whether the present authors are doing it wrong, or (c) whether the difference in point of view results from subtly different specifications of the problem. On reflection, I think the answer is (c). I

pulled out and reread the two most statistical papers the authors cite, references 15 and 17. [15] is a summary of BMA written by and for statisticians and refers to the most common application of this technology - the choice among statistical models in a context such as regression. In this context, the problem is usually formulated in a way that assumes only one of the considered models can be correct. There's a technical issue here, in that it's quite common to apply BMA in contexts where one model may be nested inside another (i.e. the former derived from the latter by setting certain parameters to 0), but in Bayesian analysis with continuous priors, the posterior probability that a particular parameter is exactly 0 will always be 0, so the models are effectively disjoint. In this context I don't think there is any argument with the traditional interpretation of BMA. However, paper [17] takes a different approach - in the context of weather forecasting rather than long-term climate, but I don't think that affects the statistical issue under discussion - they do assume there are different physical forecasting models and one of them is "best" though without ever defining precisely (I don't think) what best actually means. So this looks like exactly the kind of BMA that the present authors are criticizing. But I think it comes down to how you define the problem. The present authors define a climate model as fitting the data if it satisfies a tolerance condition. I have issues with how that condition is defined, but I'll come to that later - my basic point is that, the way the authors set up the problem, it obviously does NOT exclude the possibility that more than one model may be correct, so a BMA that assumes the contrary would obviously be wrong. That seems to me the fundamental issue, which the authors may or may not have clear in their own minds, but they have certainly not articulated it clearly in the paper.

The main philosophical question stemming from the reviewer's comment is whether the assumption of model exclusivity is appropriate when making multi-model predictions. If it is, then BMA in its present form is appropriate as a projection tool. If not, then it needs to be modified by relaxing this assumption as is done here.

There are two main arguments suggesting that the assumption of exclusivity is inappropriate.

First, exclusivity means that the conditional probability of more than one model given the observations (or future climate) is 0: $p(M_i, M_j|Y) = p(M_i|M_j, Y)p(M_j|Y) = 0$, which essentially implies $p(M_i|M_j, Y) = 0$ [unless M_j is completely excluded by the observations and $p(M_j|Y) = 0$]. This is untrue when the models are dependent in the general sense of the term. Consider, for example close-cousin models, such as IPSL-CM5A-LR and IPSL-CM5B-LR. Defining probability as degree of belief, a reasonable climate scientist would put a high conditional probability in IPSL-CM5A-LR being a reasonable model if s/he were informed that its cousin IPSL-CM5B-LR is consistent with observations, and not zero as implied by the exclusivity assumption. We add a short discussion to this extent on lines 110-118.

Furthermore, we pulled out several papers using BMA and found that due to internal variability there can be a considerable overlap in pdfs produced by different models. Indeed, this is very evident in the Raftery et al. (2005) work in Figure 3: there is almost complete overlap in the predictive pdf between two of

the highly-weighted models suggesting that many climates are jointly consistent with these two models. In fact, each model's pdf in their Figure 3 has some form of overlap with other models, due to the large internal variability and/or model error in capturing the 48-h surface temperature at Packwood, WA. Duan et al. (2007) do not show the predictive distributions for each of the candidate models, yet all models get considerable probability. There is nothing to suggest that one model being consistent with observations should preclude any other from doing so. In Figures 2 and 6 of Bhat et al. (2011) we see modeled output overlapping between climate models due to large internal variability, and many models simulate overlapping pdfs, according to their Figure 7. In Figure 2 of Olson et al. (2016) we see many models producing similar temperature output, and for future projections we see smooth pdfs stretching across temperature projection space (instead of jagged pdfs centered on individual models). Clear overlap in modelled pdfs is also evident in Figures 3 and 4 of Terando et al. (2012) who work on projections of agro-climate indices. This supports the idea that several models may be jointly consistent with a given climate. Thus, it behooves us to relax the assumption of model exclusivity.

Finally, we show the histogram of the number of models which are jointly correct as part of the Arctic ice-free year projection method (Supplementary Figure 8). The results show that only a minority of the parameter values (approximately 800000 chain members out of the 1960000-member chain) are associated with a single correct model. The most likely case is that at least two or more models are correct, highlighting the importance of incorporating non-exclusivity.

2. I also feel the authors have not clearly separated the issues of model independence and model disjointness (or exclusivity). Granted, nowhere in the paper did the authors directly confuse these two concepts (which I would describe as an elementary beginner's error of the sort I encounter when teaching freshmen) but I still feel that they have confused them in some big-picture sense. To be specific: the first two paragraphs on page 4 are clearly about model dependence, but in the third, the emphasis switches without explanation to model exclusivity. Granted, the solution the authors are proposing would be simpler if the models were independent, but that doesn't seem to me the main point of the paper.

We apologize for this confusion. Indeed, there is also confusion in the literature about the definition of model dependence. We now statistically define both concepts, which makes a clear distinction between them (see lines 99-109). The proposed method makes neither the model independence nor the exclusivity assumptions.

I have a whole host of minor issues about the paper, detailed below, but these seem to me the two major issues. I do believe the paper is making interesting points and could be rewritten in a way that would make it suitable for publication, but as things stand, I don't think it's making a clear-cut argument.

Here are the rest of my points about the paper:

3. Section 3.2 is about something called the inclusion-exclusion principle, but the authors never call it that. I can't really believe the authors have never heard that expression, but there's a wikipedia article about it if you care to look it up. So why this rather long-winded statement and even more long-winded proof (in the Supplementary Information section)? (To be explicit, the IE-principle is applied on the probability space of events that are intersected with the event A and conditioned on the random variable Y, but on that space, it's just the basic IE formula.)

Yes, the formula (7) is derived from the IE formula for probability. We now state that both in the main text (lines 143-144) and in the supplement. However, we believe that the formula is sufficiently different from the IE formula to require a separate proof. Since this is one of the crucial contributions of this paper, the formula and proof illustrates how the standard BMA can be used inappropriately.

4. However, as far as I can tell, formula (7) is never actually applied as given. Instead the authors immediately segue to equation (8), which is a much more familiar (and natural) formulation of the posterior distribution of A. So I'm still unclear about the purpose of the formula (7).

We show formula (7) to illustrate how the proposed approach (Equation 8) relates to the common methodology of Bayesian model averaging, and to show that the new approach accounts for model dependencies of the order n .

5. Section 3.4. Here my confusion was less about big-picture issues than the specifics of how the model is defined. What exactly is μ_i ? It seems it could be a parameter of dimension m for any m (line 154) and equation (8) makes else if the plus-minus operations are defined coordinatewise, but it's still not clear to me what physical quantity the parameter actually represents. Specifically, is it the 26-dimensional vector of annual means, or is it just a scalar - as much of the subsequent discussion, including figure 1, seems to suggest? It's quite common in this field of research to summarize a multi-year vector in terms of a single overall mean (even though the annual mean may be varying) and that would be by far the simplest interpretation of what the authors are doing, that also avoids the issue of whether sub-decadal effects such as El Ninos could bias the results. (That's why much of the climate literature uses 30-year means.) However it becomes clear later, this isn't the intended interpretation.

We define μ_i on lines 177-179: "annual timeseries of [...] (SIE) from 31 CMIP5 climate models y_i with unknown random means μ_i ". So, μ_i is just a constant for each model i . We add to line 179 that i is model index.

6. I also have a concern about the same notation (μ and μ^*) being used to describe physically different quantities - in one case, the proportion of sea ice, in the other, the year in which sea ice reaches a certain limit. It seems unnatural to me that one would use essentially the same statistical formulation to describe such different quantities. Maybe the authors think it is OK, but they haven't said much to justify it.

We apologize for the confusion. We changed the notation, and now refer to the future predictor variable from the i th model as z_i^* , and to the “true” future climate by z_i .

7. Page 10, equation (9). Now I'm getting REALLY confused. Why does this equation makes sense at all? The authors say they used the same idea in ref. [23] - well, I downloaded and tried to read that reference, but I couldn't see any equation that looked like (9). It would make sense to me if the authors simply defined a Delta that represented some qualitative notion of model equivalence (I'm not sure what that would be, but based on Figure 1 it looks as though a Delta of something like 0.5 on the scale of the x-axis would be reasonable) but why give it a half-normal prior distribution? And why define sigma as the "standard deviation of differences between each model's present-day mean and next-closest model's mean"? In other words, to define sigma for a single model you first have to look at ALL the other models (to determine which one is closest) and then define a standard deviation based on model differences? Please! Let's at least have a definition that allows me to determine, for ONE model and ONE set of observations, whether they are a good enough fit or not!

Here, the reviewer makes several important points to which we will attempt to respond one-by one.

First, to simplify the paper we no longer cite reference 23.

We appreciate the reviewer's idea to use a constant value for the Deltas. In fact, we have originally taken this approach. However, this produced pdfs which were flat for the dependence method (“New” line in Figure 1). This is because, under such assumptions, all future years within an assumed tolerance of a correct model are possible, but all other years are excluded, with an abrupt cutoff at Δ^* . There is no reason our uncertainty about the future climate should abruptly cut-off at a given fixed year. Thus, fixing the values of Deltas appears to be inadequate in terms of accounting for the uncertainty inherent in Deltas.

Since Δ and Δ^* may have an influence on the outcome, and we do not have any reasonable way of choosing their values, the most reasonable way is to obtain them via the Bayesian approach of treating them as unknown parameters and estimate them via the data. Here, all models use the same present-day and future Deltas.

The reviewer mentions a desire to have, for a single model, and a single set of observations, a tolerance measure that allows to determine whether they are a good enough fit or not. We believe that to make the hypothesis combination fair, the tolerance has to be the same for all models. Otherwise, for the same difference from observations, one hypothesis (model) would be considered correct, and the other is not.

Figure 1: Pdfs for the Arctic ice-free year obtained from the original version of the code with fixed Δ and Δ^* . Blue: dependence method, red: standard BMA. Vertical solid lines: means and 90% posterior credible intervals.

The reviewer also questions why we have chosen this particular form for tolerance prior: $N^+(0, [f\sigma]^2)$. First, we choose the half-normal distribution because the tolerances cannot be negative, and because it results in reasonably-looking posterior distributions, and weights that gradually taper off further away from the observations. We now specify this in text on lines 207-209. In choosing the standard deviation for the prior, we were guided by the fact that the tolerance should broadly depend on the degree of model error in the ensemble. One such measure of error is next-closest model differences. An explanation for this is provided by the leave-one-out cross-validation framework: if one model is assumed to be true climate, then the difference between the next closest model and the “truth” provides a sample of model error for the “best” model. Repeating the procedure for each model, we get next-closest model differences. Here, we calculate, for each model, the difference of SIE to the next closest model, and then take a standard deviation of these values between all models, to obtain σ , which we use for the tolerance prior. Thus an estimate of σ obtained in this way gives us an idea about the variability of the minimum discrepancy between models. Similar parametrization is used for future tolerance Δ^* . Both standard deviations are then post-multiplied by a correction factor f to account for potential overconfidence (e.g., all models may have a common bias, thus considering unscaled inter-model differences may result in tolerances which are too small).

Other ways of defining tolerance priors may be possible. For example, average model-observational distances may provide another option to parametrize the prior, instead of σ . However, in that case, only the present-day tolerance can be calculated, since there are no observations for the future. This is the reason we chose a purely model-driven σ . Yet another option may be to use model range instead of σ . However, such metric may be heavily influenced by one poor outlier model. At any rate, for the Arctic ice-free year experiment, the relative sizes of ranges are similar to the relative size of standard deviation of closest-model differences: e.g., $\frac{\sigma^*}{\sigma} = 10.8$ and $\frac{\text{range}(z_i^*)}{\text{range}(\mu_i)} = 13.0$. Thus, we believe that choosing one method over another will likely have a small impact on the results.

We have modified the text accordingly to defend our choice of the priors on lines 202-211. We also stress in the caveats paragraph that testing other prior distributions is left to future work on lines 474-478. We believe these changes have substantially improved the manuscript.

8. Equation (10). Here I have an issue about the assumed form of prior distribution for the μ 's and Δ 's and, especially, whether this is consistent with the notion of model dependence, but some of this is discussed subsequently. See also the top of page 11.

Using independent priors for μ_i 's does not imply independence of the μ_i 's in the posterior distributions, the data informs on the amount of dependence. This is a common practice in Bayesian statistics.

9. Line 201: given the discussion at the beginning of this review, isn't the discussion of "standard BMA" just a straw-man comparison? In other words, standard BMA assuming disjointness may be justified when the hypotheses really are disjoint, but when they are not, isn't this just wrong?

In comparing the new methodology to a standard BMA we merely follow the standard procedure in our discipline of comparing a new method to a previous one.

10. Line 219 - this seems to be the first time in the whole paper that the acronym AMOC is used, so the authors should define it. (Atlantic Meridional Overturning Circulation, as I remembered when I looked at ref. 23.)

We thank the reviewer for pointing this out. We now define this acronym.

11. I don't understand lines 223-232 at all. Just rescaling the prior density doesn't change the posterior, because the posterior density always has to integrate to 1. So why do this? The authors themselves say "this weighting approach is incorrect..."

We apologize for this confusion. We have modified the notation to indicate that the number of correct models m in this additional experiment is a function of model parameters (z^*, μ) (see lines 307-310). The main purpose of this section

is to show that such priors (up-weighting the converging models) is wrong, and thus to provide additional support for the default priors used for the main model experiments.

12. Lines 246-251. I'm not sure that the authors are justified in assuming that every instance in which two models produce similar results has a causal explanation ("subtler sharing of ideas", etc.). They could just be random chance. However, there might be a possible statistical test whether the number of near-coincidences in the model projections is greater than could be explained assuming model independence.

In fact, we already state that similarity may be caused by a random realization of internal variability, see lines 334-335. We do, however, add equifinality to the list of reasons (different parameters and modelling structures resulting in the same output by chance), see lines 332-333.

13. Section 3.6. The results are interesting but hardly earth-shattering. The key point, obviously, is how one interprets the anomalous projections of the model FIO-ESM. I would actually have appreciated more discussion of the physical reasons for this, e.g. whether the explanation offered in lines 215-216 is something that physicists consider important. As it stands, compared with what the authors call "standard BMA" (though I'm not sure the word standard is appropriate, for reasons explained earlier in this review) the 95th percentile of the distribution is postponed 8 years. But the lower tail is hardly changed at all, and for people worried about the pace of climate change, that is surely the more important end of the distribution.

We believe the change of the probabilistic projections of Arctic ice-free year due to heavier up-weighting of the exclusive FIO-ESM model is just one conclusion of the paper, with two other important statistical conclusions being (i) a new method to find probabilities for non-exclusive hypotheses and (ii) a method to provide probabilistic projections under such hypotheses, accounting for all levels of inter-hypothesis dependence.

While we agree with the reviewer that it is important to figure out why this model differs from other models, we believe a comprehensive analysis of FIO-ESM physics is beyond the scope of this paper. We thus leave it to future work (see new text on lines 301-303). We do, however, add a discussion on the skill of FIO-ESM at simulating several present-day Arctic ice parameters, based on previous work by Shu et al. (2015) (the paragraph starting at line 293).

14. Line 326: this is the first explicit acknowledgement that the authors are using a time series approach. They really need to carefully define all their assumptions at the beginning of the paper, not near the end.

In fact, we do state that we work with the time series data at the beginning of the exposition of our approach on line 177.

15. Line 363: again, the first time that trend, autocorrelation and standard deviation (of the individual data values) have been mentioned.

In fact, we already mention that we work with autocorrelated time series on line 177. However, we now also explicitly mention that we assume the AR(1) structure and that all parameters other than the means are fixed on lines 185-189.

16. Line 373: do I understand from this that the decision to end in 2004 was made deliberately, to avoid the increase of natural circulation variability? If so, does that bias the results in any way?

One of the reasons for ending the year in 2004 was that it is the end of the historical period in the CMIP5 simulations. The output after year 2004 no longer uses observed radiative forcings, but rather relies on their simulated projections. Hence, any model weights that result from post-2004 observational constraints can, in principle, depend on the future projection scenario used (RCP8.5 vs RCP4.5). Addressing the concerns of the reviewer, we perform sensitivity analysis using data until year 2017 and do not find any critical differences in the projection pdfs. The Arctic ice-free year pdf figure (Figure 4) now includes the sensitivity analysis. We have also performed the same sensitivity analysis for temperature change to melt, a new projection variable requested by Reviewer 3.

17. Line 385 - "removes the spurious mid-21st century .. peak ..." - there's still a bimodality, but it's less pronounced under the new approach.

We slightly changed to wording to say "removes most of the spurious mid-21st century peak". The bimodality generally refers to existence of two distinct peaks in the pdf over the entire range of a random variable.

18. Equation (13). As far as I can tell, these equations are just wrong. Or if you think they are correct, at least say what the formulas actually are (i.e. give explicit algebraic expressions). If you have an AR(1) model with trend, then the four parameters of the model are the autocorrelation, the innovations SD, the mean and the trend. Any textbook on time series analysis will give you the joint density of the observations under this model. But it doesn't look anything like (13).

We apologize for being too curt in our equations. We have added a large amount of material in the "Statistical Model and Detailed Likelihood Function" subsection that shows in detail our statistical model, as well as the formulas for the relevant likelihood functions (lines 531-544 and 555-559).

19. Pages 20-21: I would have appreciated more discussion of the rationale behind the length of MCMC runs, including the burn-in. The authors use a much longer MCMC for the future projections part of the experiment than the past observations. Maybe they reached this conclusion after sound statistical testing, but it's not clear to me. Also, the burn-ins are much shorter than I would have expected - modern practice of MCMC often uses burn-ins of a quarter or even half of the entire MCMC run. Did the authors test the sensitivity of the results to this assumption?

We thank the reviewer for this important point. We use shorter MCMC chain length for observation system simulation experiments partly for computational

reasons, since these experiments essentially run the code 31 times – once for each model, and partly because these experiments are not used for the projections. We tested whether these shorter chains (and the associated burn-ins) are reasonable. To do so we implemented an ice-free year real estimation run with chain length of 150,000 and a burn-in of 10,000 as for the OSSE experiments. We compared pdfs for all parameters in this run to the longer run used in the main manuscript. The results do not indicate any considerable mis-convergence (Figure 2), suggesting that the shorter chain length used for the OSSEs is reasonable. Specifically, the pdfs for z^* (the only parameter relevant for confidence interval validation) appear to be reasonably similar.

Figure 2: Comparison of pdfs for all estimated parameters between a shorter real estimation run and the one used for the paper. The 95% posterior credible intervals are shown for the shorter run.

In addition we consider the chain plot for the short real estimation run. The chains (with the burn-in discarded) are well-mixed and do not appear to be unduly influenced by the initial random seed (Figure 3). Thus, since the OSSE

runs are not used for future projections, we consider this shorter chain length acceptable for the method calibration purposes.

Figure 3: Chain plot for the short estimation run for the Arctic ice-free year projections. Subplots refer to parameters. In each subplot the x-axis reflects the position of the parameter within the chain.

The expanded discussion on the MCMC settings for the observational system simulation experiments is shown on lines 569-575.

20. The Supplementary Information was submitted in a form that made it very difficult to read. The figures were submitted as seven separate postscript files but there is no labelling which is which, and several did not display correctly in the postscript viewer I use (GSView). Why can't you convert to pdf and submit the entire SI as a single pdf file, just like the main paper?

We considered the reviewer's suggestion and have now put together a single pdf file with all the supplementary material. We believe this change has made the supplementary material more orderly and easily accessible.

References:

Bhat, K. S., Haran, M., Terando, A., & Keller, K. (2011). Climate Projections Using Bayesian Model Averaging and Space-Time Dependence. *Journal of Agricultural, Biological, and Environmental Statistics*, 16(4), 606–628. <https://doi.org/10.1007/s13253-011-0069-3>

Duan, Q., Ajami, N. K., Gao, X., & Sorooshian, S. (2007). Multi-model ensemble hydrologic prediction using Bayesian model averaging. *Advances in Water Resources*, 30(5), 1371–1386. <https://doi.org/10.1016/j.advwatres.2006.11.014>

Olson, R., Fan, Y., & Evans, J. P. (2016). A simple method for Bayesian model averaging of regional climate model projections: Application to southeast Australian temperatures. *Geophysical Research Letters*, 43(14), 2016GL069704. <https://doi.org/10.1002/2016GL069704>

Raftery, A. E., Gneiting, T., Balabdaoui, F., & Polakowski, M. (2005). Using Bayesian model averaging to calibrate forecast ensembles. *Monthly Weather Review*, 133(5), 1155–1174. <https://doi.org/10.1175/MWR2906.1>

Shu, Q., Song, Z., & Qiao, F. (2015). Assessment of sea ice simulations in the CMIP5 models. *Cryosphere*, 9(1), 399–409. <https://doi.org/10.5194/tc-9-399-2015>

Terando, A., Keller, K., & Easterling, W. E. (2012). Probabilistic projections of agro-climate indices in North America. *Journal of Geophysical Research Atmospheres*, 117(8). <https://doi.org/10.1029/2012JD017436>

Response to Reviewer 3

In this article, the authors present a novel approach to account for model dependencies in multi-model ensemble analyses. This topic is relevant for the community as it has been shown that ignoring model dependence might lead to an overestimation of some results while others might be ignored even though they are more correct. The authors develop a new method based on the Bayesian model averaging.

The focus of this review is not on the statistical correctness or novelty of the approach presented here but on the results concerning the sea ice projections. Thus, in the following it is assumed that the statistical method is robust and the following points refer mainly to the sections 3.4-3.6.

We thank the reviewer for their thoughtful review of the manuscript. We respond to the reviewer queries by adding the analysis performed for a different variable (global mean surface temperature [GMST] change at which September Arctic ice will effectively disappear under the RCP8.5 scenario “GMST change to melt”), by performing the sensitivity analysis to both the time period and the observational dataset for historical Arctic sea ice, and also by additional explanations and discussion. In addition, we re-ran all the main experiments used in the original manuscript to eliminate potential code bugs. There are no changes except that the exclusive BMA 95% percentile for Arctic ice-free year goes up to year 2085 (instead of 2084). We believe the reviewer’s comments and our revision has substantially improved the quality of the manuscript. Our responses to the reviewer are marked by bold blue in this file, and using track changes in the main manuscript. All line references are for the manuscript version with tracked changes.

Major Comments

1. My main concern is the fact that you look at the year when the Arctic will become ice-free. In my point of view it is more interesting to get to know the level of warming for an ice-free Arctic and not the year as this is dependent from the underlying emission scenario. The driving force for sea-ice evolution is given by temperature (or the amount of global warming) and thus by the emissions. It should be possible to use this approach in a similar way to calculate the pdf for global warming (and/or for CO2 emission)!? This could give much more information than your approach now.

We think the reviewer’s suggestion is very useful. We have repeated the analysis on the GMST change to melt. Note that this analysis results in a new set of individual and joint weights. For simplicity, we show weights only for the Arctic ice-free year experiment, while the results for the pdfs are now shown for both variables in Figure 4 of the main manuscript. The abstract and the discussion have been modified to incorporate the new results. Note that this GMST change is conditional on the RCP8.5 scenario, and the real-world GMST change at which the ice will effectively disappear is likely sensitive to the trajectory of the emissions. Hence, we do not believe we can make a general statement about the probability of ice disappearing at a certain GMST change. To do that we need to analyze temperature stabilization experiments. This may be an interesting idea for future research, but is beyond the scope of this work.

2. How sensitive are your results on the choice of the time period for present-day? For me it is not intuitive to stop at 2004 (even though the historical simulations end in 2005). (Maybe present-day is just a wrong formulation.) How sensitive are the results to your choice of observational datasets? It has been shown that the observations differ largely (see [32]).

As requested by the reviewer, we have performed sensitivity analysis to both the dataset length and the observational source. Specifically, we have repeated the analysis for both Arctic ice-free year and the GMST change for the period 1979-2017 (using a newer version of the Arctic sea-ice index), as well as using the HadISST1-derived SIE for years 1979-2017. The results for the Arctic ice-free year are only marginally sensitive to the period and the dataset. There is slightly more sensitivity in case of the GMST change, mainly affecting the strength of the secondary peak between 3.5 and 4 °K.

3. The joint hypothesis is defined as correct if all constituent hypotheses are correct (ll.187-...) - would it be possible to define it in a different way? It could be that two h_i itself are wrong but together they are correct. Maybe a sentence about a possible expansion of this assumption could be added.

In standard logic theory, A and B means that both A and B are true. Hence, it is not possible to define joint hypotheses in a different way.

4. Supplementary Figure 1: Could you please explain the local minima in some of the pdfs. For me it is not clear why the pdfs look like this – further investigations could be made when forming groups of models with similar behavior and discuss the shape of the pdf. Do you have an idea of the underlying reasons for differing that largely? What is the difference of the underlying “true” models?

In many cases, the pdfs in Supplementary Figure 1 have some resemblance to the pdf obtained using real observations: there is a broad plateau or a large peak over the first half of the century, and a secondary peak during the late century. If we select any model (except the exclusive FIO-ESM model) with present-day SIE roughly between 6 and 7 million km² as pseudo-observations (imagine replacing one of colored circles in Figure 4 of the main manuscript with a vertical red line) then all other models close to that line will receive considerable weight. These models will generally form a large cluster over the first part of the century, which can explain broad pdf peaks over this period. In addition, the exclusive FIO-ESM model also simulates similar present-day SIE, meaning that it would get some weight. However, this model produces very late ice melting. This would explain the secondary peak around year 2090.

Other types of pdfs can result if we select models with extreme low/high present-day SIE as the “truth”.

Since these pdfs are not relevant for future projections, and are only meant to demonstrate that the method is relatively well calibrated, we refrain from any discussion of the pdf shapes in the body of the manuscript.

5. In paragraph 3.5, you discuss about the behavior of the model FIO-ESM because this is simulating present-day SIE well but loses ice much later than the other models. Could you say something about the sea ice thickness? As I understood you only look at the scenario RCP8.5, thus I would delete the sentence about temperature changes under RCP2.6 as this does not give an explanation. The explanation is given in the next sentence (ll.219-222).

We thank the reviewer for bringing the up. We found a reference (Shu et al., 2015) comparing the performance of FIO-ESM to observations and reanalyses, including the reanalysis of Arctic sea ice volume (SIV), which is a measure dependent on sea ice thickness. FIO-ESM has reasonable performance as assessed by the monthly SIV and SIE root mean square error. FIO-ESM mean annual SIV, its seasonal cycle, and the trend are within a reasonable distance from the reanalysis. However FIO-ESM has much larger SIV variability compared to the reanalysis, and its monthly linear SIE trend is almost half of the observations. Overall, FIO-ESM performance in capturing Arctic sea ice properties is comparable with other models. We expand the discussion on FIO-ESM performance on lines 293-303.

As for the cooling in the Northern Hemisphere in FIO-ESM, it is defined as the temperature difference between the model version with the surface wave effects, and without. Hence, we believe the results are general rather than scenario-dependent. Thus, we retain the sentence about the temperature changes while removing reference to the scenario (see lines 286-288).

6. What do you mean with “model convergence” (l.229)?

Here, by model convergence we mean the number of models per area of model output space. We change the text to reflect that on lines 312-313.

7. ll. 362-367 Could you give a suggestion on how to include these uncertainties and by how much the computational costs would increase (I don't want to see any numbers, but I have no feeling on how large the increase factor would be and how complicated an expansion would be...).

The uncertainties in each model's slope, innovation standard deviation, and autocorrelation can be included in the MCMC as additional parameters. This can be achieved by introducing for each model i an uncertain slope parameter k_i , autocorrelation parameter ρ_i , and innovation standard deviation parameter σ_i . This would increase the total number of parameters to 128. While there are no theoretical constraints on running the MCMC with such a number of parameters, we have been running into memory problems in R programming language while increasing our chain length beyond the current size of 2 million with just 35 parameters. Using more parameters may also lead to slower convergence, requiring even longer chains.

8. ll.373-374 How would that change the result? It could reduce the observational uncertainty to use longer time-series (until 2017, instead of 2004 only), but would include the trend. (See comment 2.)

See our reply to query #2. Note that we have not considered an uncertainty in the trend for this sensitivity analysis. As previous work has found, the Arctic ice-free year is more strongly correlated with mean historical September SIE compared to September SIE trend, SIV, and other metrics in the CMIP5 models (see lines 214-219). We leave considering the trend uncertainty to future work (line 465).

Minor comments:

II. 20-21 a reference is missing for “postpone” - if you say the new year it would be nice to get the information on how many years the mean or the 95th percentile of the pdf is shifted

Thank you for this suggestion. The 95th percentile shifts from year 2085 to 2092. We now explicitly stipulate this in the Abstract.

I.58 statistical meaningful is a very unprecise formulation- could you justify this somehow?

As part of the editing process, the offending term has been removed. We now state that we first define mathematically terms “model dependence” and “model exclusivity” on lines 57-58.

I.149 delete “the”

Done.

II.155-... could you specify the “uncertain random tolerance ranges”; what are your numbers for f and σ and Δ ? - you could maybe include a table in the supplemental material.

Thank you for your query. We provide the f values on lines 576 and 582. We now also specify $f\sigma$ and $f\sigma^*$ on lines 211 and 213, respectively (we believe these values are more useful to the reader than the standard deviations themselves). In addition, we provide a new paragraph related to the tolerances and their pdfs on lines 264-274.

Supplementary Figure 2 the one-to-one line would be helpful to easily see the relationship.

Thank you, we have now added the 1:1 lines to this plot, and to the similar plot for the temperature change to melt (Supplementary Figures 2 and 4).

I.240 “present-day” instead of “present day”

This has now been corrected.

Fig.3 What do the colors mean in Fig.3?

The colors are added to beautify the plot. They have no meaning. We now state this in the plot caption.

I.262-264: How do I see that in Supplementary Fig.5?

As the figure shows, there are only about 800,000 MCMC samples corresponding to just one correct model, while the total number of MCMC samples is 1,960,000. Thus, the probability of only one model correct is less than 50 %. We have moved the figure reference to after the end of the sentence “Specifically, only about 800,000 of 1,960,000 MCMC samples correspond to just one correct model” (lines 349-351).

I.336 “it” instead of “is”

We have corrected the typo.

I.390 write “[11]” instead of “Sanderson et al. (2015)”

References:

Shu, Q., Song, Z., & Qiao, F. (2015). Assessment of sea ice simulations in the CMIP5 models. *Cryosphere*, 9(1), 399–409. <https://doi.org/10.5194/tc-9-399-2015>

Reviewers' comments:

Reviewer #2 (Remarks to the Author):

Review of second version of Olson et al.

This is Reviewer 2 of the original version.

The authors have substantially revised the paper in response to my and the other reviewer's comments and have also provided a full response to both reviews. I particularly appreciated the very detailed responses to my own comments. Undoubtedly the paper is much improved. I still feel there are some unclear points and ambiguities, and a few new points have arisen, so that the paper requires some further revision in my view, though I think it's a lot closer to ultimate acceptance now.

The response to my review: the only point that the authors responded to that I'd like to pursue further now is my original point 7, specifically why the authors adopted a random rather than deterministic view about the Delta constants that arise in defining model fit.

The authors' response is that the assumption of common Deltas leads to sharp cutoffs for the year that a correct model is deemed to fit the data, which I agree is an undesirable property. Nevertheless I feel that my original point of view was a reasonable one to start out with, and indeed the authors state that they also started out with this assumption. I believe this is a sufficiently subtle point that it deserves to be given explicitly in the paper! In other words, acknowledge that a fixed Delta at first sight seems more natural, explain why it led to difficulties, and use that to motivate the actual model with a random Delta.

I read the revised form of the paper from scratch and I still have quite a few points to make:

1. Lines 163-4: This may be more a stylistic than a technical point, but I winced a little when I read "the rest of this section can be skipped by non-statisticians..." (as a statistician myself) since that implies that the authors have their own doubts about the importance of including this material. I think a modest change of wording would avoid that implication.

2. Lines 176-178: Thank you for the more precise description of an AR(1) model (I am fine with this version) but why did the authors treat the other parameters (apart from the linear regression parameter) as fixed? Since they now do have an explicit likelihood function that includes the first-order autocorrelation and the residual variance, there is no problem in principle with treating these parameters as unknown and putting prior distributions on them for a Bayesian analysis. There is, as far as I can see, no physical justification for assuming these parameters to be known. However there could be technical complications, for example if one had to augment Gibbs sampling with a much more slowly converging Metropolis algorithm, or maybe the posterior sample paths (under an expanded model that treats these parameters as unknown) wander too much over the parameter space for the results to be easily interpreted. Or maybe the authors just gave up on trying to implement that step. Whatever the true reason, I feel there should have been some more detailed justification of this step, since from a classical statistical theory viewpoint, it is always somewhat problematic to treat parameters as known in advance when they are not.

3. Top half of page 11: this is where I feel the paper needs more explanation of the authors' approach to tolerances, see my intro comments above.

4. Lines 197-198: Where do the values 0.79, 8.5 come from?

5. line 199, hypothesis rather than hypotheses (maybe the authors disagree, but this appears more consistent with subsequent usage, e.g. line 224)

6. Equations (11): the chain of integral identities needs some further explanation I feel. In line 2, $p(Y|\mu)$ should really be written $p(Y|z^*, \mu, \Delta, \Delta^*)$ to match the notation of line 1, but I

presume the explanation there is that once μ is specified, the other parameters are irrelevant to the distribution of Y . I had a harder time understanding the third line - what happened to z^{**} here? Possibly the intention is that once the Deltas are defined, the prior density of z^{**} is assumed flat over the region on which it is positive, but this wasn't obvious to me based on the explanations in the paper up to this point. In any case, I do feel the readers deserve more explanation.

7. Lines 288-299: Would this paragraph be better omitted? As any Bayesian statistician will tell you, it's always doubtful when someone defines a prior distribution based on empirical estimates computed from the data, though there are instances in Bayesian statistics when such methods have been proposed and carefully defended. However, in this case, the authors also question the validity of the method (lines 296-297). So perhaps leave it out altogether?

8. Lines 444-447: these lines relate to my earlier comment about assuming some of the parameters in the AR(1) model known. I don't quite see what the "potential identifiability issues" are, though I do understand why the authors are concerned about the "computational cost and complexity" of doing this. My overall comment is that the justification for this step needs to be clear, though if "computational cost and complexity" are the real reason, maybe they should just say so and leave it at that.

9. Lines 455-458: back to the issue of random tolerances. These two sentences seem to me an attempt to express very succinctly the point they spelled out at much greater length in their response to my review, but I have already expressed the viewpoint that it needs a more detailed explanation in the paper to be intelligible.

Overall, my feeling is the paper still needs another round of revision, but hopefully they will find the points in my present report ones that they can respond to more easily and directly than with my first review of this article.

Reviewer #4 (Remarks to the Author):

I have been asked to review the revised version because Reviewer 3, who focused on the sea-ice aspects of this paper, is on extended leave. As such, I aim at aligning the review in the following along the lines of the previous review in as far as possible. However, this is unfortunately not always possible.

In the following, parallel to reviewer 3, I only comment on the sea-ice related aspects of this study, assuming that the underlying statistical method is correct. Doing so, I strongly second the major concern of the previous reviewer that the focus on a specific year of an ice-free Arctic Ocean is misleading. Indeed, I recommend leaving out this discussion altogether. There are several reasons for this assessment:

First, time is not a driver of the state evolution of the system. Hence, any reference to a specific year will always depend on the trajectory of future emissions which is highly uncertain. Focusing instead on the actual drivers of the system, be it temperature or CO₂ emissions, allows one to obtain a trajectory-independent answer as to the conditions under which the Arctic might become sea-ice free in summer.

Second, in contrast to the reply by the authors to the previous reviewer's comments, the real-world GMST at which the ice will effectively disappear is barely sensitivity to the trajectory of future emissions or future temperature evolution. This is among others shown by the very robust linear relationship between global-mean or Arctic temperature and Arctic sea-ice coverage that is robust in all CMCIP5 models and the observations, and also by the linear relationship between emissions and Arctic sea-ice coverage, which again is robust in all CMIP5 models and the observational record independent of trajectory (see, for example, Notz and Stroeve, 2016, <https://doi.org/10.1126%2Fscience.aag2345>). In contrast, the sea-ice evolution is by no means a linear function of time as apparently implied here.

I hence question the validity of the analysis carried out here regarding the future evolution of sea ice as a function of time. It is obvious from physical principles that the two main control variables on the future evolution of the ice cover are (a) how much ice is there just now (which explains the correlation with SIE examined by the authors), and (b) how fast does the ice disappear (the "correctness" of which in individual models is apparently neglected here). As pointed out for example by Rosenblum and Eisenman (2017, <https://doi.org/10.1175%2FJCLI-D-16-0455.1>) or Notz and Stroeve (2016) almost all CMIP5 models have too low a sensitivity of past and future ice loss, so any study neglecting the impact of the sensitivity will most likely underestimate the speed of future sea-ice loss. Note, in addition, that Stroeve and Notz (2016, <https://doi.org/10.1016%2Fj.gloplacha.2015.10.011>) showed that reasonable past performance is no indication for reasonable future performance.

Ignoring the details of the statistical analysis, I believe that the following two questions must robustly be answered before the method itself can be applied:

1. Do we capture all relevant parameters of "model quality" when weighting the "correctness" of any given model?
2. Are our assumption on the physical relationship between our control variable (here: time) and the driven parameter (here SIE) robust?

Both questions can probably be much easier by answered by "yes" if one examines sea-ice evolution as a function of either temperature or emissions, including a simultaneous assessment of the quality of current SIE and of the models' sensitivity to warming or to emissions.

I fear that the method might erroneously be judged as wrong if the paper were published in the present form, because we know with great certainty that the estimated PDFs of when the Arctic sea ice might be gone are erroneous. You will be hard pushed to find any sea-ice scientist who could even remotely imagine to have an Arctic sea-ice cover to exist beyond 2060 in an RCP8.5 scenario...

As it stands, I would trust results from carefully bias corrected model simulations (e.g., reference [31], or Sigmond et al., 2018, <https://doi.org/10.1038%2Fs41558-018-0124-y>), or from studies which exploit the linearity between forcing and sea ice (ref. 32), far more than I would trust a fairly different result from the approach taken here because I believe that the input data used here, and the underlying assumptions on physical relationships, are not robust. Hence, if the authors find very different ranges of future sea-ice evolution than other recent studies, these differences should be explained based on our physical understanding of the system.

Other issues I noted:

There is somewhat misleading notion of a possible "model correctness" in the current version of the paper. However, we know that all our climate models are wrong - "and not even approximately true". (Parker, 2009, <https://doi.org/10.1111%2Fj.1467-8349.2009.00180.x>). They can nevertheless be very useful to answer specific questions, which is what we use them for. It would be helpful to have this line of thinking reflected in this paper.

Recently, Olonscheck and Notz (2017, <https://doi.org/10.1175%2FJCLI-D-16-0428.1>) established a notion of "model consistency", which implies that the deviation of a specific model from an observational record is consistent as long as the deviation is smaller than the model's simulated internal variability (or standard deviation). They found a wide range in sea-ice variability across models, in contrast to the claim in line 204 (version with marked changes) that the standard deviation is constant across all models. Ideally, the models' different standard deviation should for the present study be estimated from the control simulation and then used for the statistical analysis for each model individually. Realistically, it should at least be noted that this should be done for a more robust assessment :-)

Figure 3: If the colors mean nothing, I strongly suggest to not vary them.

I hope the authors will find these comments helpful in re-doing the analysis of future Arctic sea-ice evolution. The method suggested here seems powerful and, if robust, would be very welcome by

the community. Using a misleading example to sell its usefulness could overshadow the beauty of the underlying statistical approach.

Best wishes.

Dirk Notz

Response to Reviewer #2

We thank the reviewer for their positive view of our latest revisions. We respond to the reviewer queries mainly by minor clarifications and changes both in this response and within the body of the paper. We note that in response to the fourth reviewer the method has changed to involve a second constraint of sea ice sensitivity to temperature, in addition to mean SIE. This necessitated some changes to the equations used, although the overall statistical framework is the same as previously. We also note that (also in response to reviewer 4) we have removed the projections for the ice-free year, and have run several additional experiments focusing on the global mean temperature (GMST) change to melt. Note that we have also re-defined all GMST changes with respect to preindustrial (as opposed to years 1960-1999), which allows us to reference the Paris agreement. All lines refer to the version of the manuscript with changes tracked.

Major Points

The authors have substantially revised the paper in response to my and the other reviewer's comments and have also provided a full response to both reviews. I particularly appreciated the very detailed responses to my own comments. Undoubtedly the paper is much improved. I still feel there are some unclear points and ambiguities, and a few new points have arisen, so that the paper requires some further revision in my view, though I think it's a lot closer to ultimate acceptance now.

The response to my review: the only point that the authors responded to that I'd like to pursue further now is my original point 7, specifically why the authors adopted a random rather than deterministic view about the Delta constants that arise in defining model fit.

The authors' response is that the assumption of common Deltas leads to sharp cutoffs for the year that a correct model is deemed to fit the data, which I agree is an undesirable property. Nevertheless I feel that my original point of view was a reasonable one to start out with, and indeed the authors state that they also started out with this assumption. I believe this is a sufficiently subtle point that it deserves to be given explicitly in the paper! In other words, acknowledge that a fixed Delta at first sight seems more natural, explain why it led to difficulties, and use that to motivate the actual model with a random Delta.

We thank the reviewer for their positive view on the revised paper. We have inserted a discussion motivating the use of the uncertain tolerances, along the lines suggested by the reviewer, on lines 215-219. We revisit this briefly again in the caveats on lines 653-656.

I read the revised form of the paper from scratch and I still have quite a few points to make:

1. Lines 163-4: This may be more a stylistic than a technical point, but I winced a little when I read "the rest of this section can be skipped by non-statisticians..." (as a statistician myself) since that implies that the authors have their own doubts about the importance of including this material. I think a modest change of wording would avoid that implication.

We agree with the reviewer. This has been now removed.

2. Lines 176-178: Thank you for the more precise description of an AR(1) model (I am fine with this version) but why did the authors treat the other parameters (apart from the linear regression parameter) as fixed? Since they now do have an explicit likelihood function that includes the first-order autocorrelation and the residual variance, there is no problem in principle with treating these parameters as unknown and putting prior distributions on them for a Bayesian analysis. There is, as far as I can see, no physical justification for assuming

these parameters to be known. However there could be technical complications, for example if one had to augment Gibbs sampling with a much more slowly converging Metropolis algorithm, or maybe the posterior sample paths (under an expanded model that treats these parameters as unknown) wander too much over the parameter space for the results to be easily interpreted. Or maybe the authors just gave up on trying to implement that step. Whatever the true reason, I feel there should have been some more detailed justification of this step, since from a classical statistical theory viewpoint, it is always somewhat problematic to treat parameters as known in advance when they are not.

We thank the reviewer for bringing this point. Using larger number of parameters can lead to slower convergence of the MCMC chains. And trying the algorithm with larger chains has resulted in computer memory problems. In addition, fixing these parameters makes exposition of the new method simpler. We thus mention “computation cost and complexity” as the reason for keeping these parameters being fixed (lines 204-205 and 633-636).

3. Top half of page 11: this is where I feel the paper needs more explanation of the authors' approach to tolerances, see my intro comments above.

Please see our reply to the intro comments above.

4. Lines 197-198: Where do the values 0.79, 8.5 come from?

As we specify in the manuscript, f is chosen so that the method gives approximately correct empirical coverage of the 90% posterior credible intervals. The standard deviations $(\sigma_\mu, \sigma_z^*, \sigma_u)$ are the standard deviations of inter-model differences in present-day model means, future GMST changes to melt, and ice sensitivities, respectively. The values in question are the products of f and the standard deviations.

5. line 199, hypothesis rather than hypotheses (maybe the authors disagree, but this appears more consistent with subsequent usage, e.g. line 224)

Thank you for this suggestion. We have changed the syntax according to the reviewer's suggestions.

6. Equations (11): the chain of integral identities needs some further explanation I feel. In line 2, $p(Y|\mu)$ should really be written $p(Y|z^*, \mu, \Delta, \Delta^*)$ to match the notation of line 1, but I presume the explanation there is that once μ is specified, the other parameters are irrelevant to the distribution of Y . I had a harder time understanding the third line - what happened to z^* here? Possibly the intention is that once the Deltas are defined, the prior density of z^* is assumed flat over the region on which it is positive, but this wasn't obvious to me based on the explanations in the paper up to this point. In any case, I do feel the readers deserve more explanation.

The reviewer is correct about the second line of the Equation 11. $P(Y)$ is independent of μ , Δ , and Δ^* so they are removed from the equation (note that the notation and the statistical model has changed somewhat to account for including a second constraint of sea ice sensitivity). To make this more clear, we have expanded Equation (11), and inserted a brief note on this on line 250.

We have also expanded Equation (11) to clarify that $p(z^*, \mu) \propto \mathbf{1}_S$, so that it is uniform over set S (at least one model being correct), and zero outside of S .

7. Lines 288-299: Would this paragraph be better omitted? As any Bayesian statistician will

tell you, it's always doubtful when someone defines a prior distribution based on empirical estimates computed from the data, though there are instances in Bayesian statistics when such methods have been proposed and carefully defended. However, in this case, the authors also question the validity of the method (lines 296-297). So perhaps leave it out altogether?

Thank you, we have removed the offending paragraph.

8. Lines 444-447: these lines relate to my earlier comment about assuming some of the parameters in the AR(1) model known. I don't quite see what the "potential identifiability issues" are, though I do understand why the authors are concerned about the "computational cost and complexity" of doing this. My overall comment is that the justification for this step needs to be clear, though if "computational cost and complexity" are the real reason, maybe they should just say so and leave it at that.

We have now removed the reference to the "potential identifiability issues".

9. Lines 455-458: back to the issue of random tolerances. These two sentences seem to me an attempt to express very succinctly the point they spelled out at much greater length in their response to my review, but I have already expressed the viewpoint that it needs a more detailed explanation in the paper to be intelligible.

We have expanded here to motivate why we have chosen to work with uncertain, rather than fixed tolerances, and now mention the issue of the flat pdfs with sharp cutoffs (lines 653-656). (See also additional discussion in the section exposing the new method on lines 215-219).

Response to Reviewer #4 (Prof. Dirk Notz)

We thank the reviewer for their critical review of the paper, focusing on the ice modelling aspects of the manuscript. We have essentially re-done most of the experiments from scratch to satisfy the reviewer's requests. Specifically, we now use an additional constraint of sea ice sensitivity to temperature in most of the experiments. In addition, we center the paper on the GMST change to melt, and remove the parts pertaining to the Arctic ice-free year. Note that we have also re-defined all GMST changes with respect to preindustrial (as opposed to years 1960-1999), which allows us to reference the Paris agreement. We respond to other queries by minor plotting changes, and by clarifications and additional discussion, both in this document, and in the manuscript body. All lines refer to the version of the manuscript with tracked changes.

I have been asked to review the revised version because Reviewer 3, who focused on the sea-ice aspects of this paper, is on extended leave. As such, I aim at aligning the review in the following along the lines of the previous review in as far as possible. However, this is unfortunately not always possible.

In the following, parallel to reviewer 3, I only comment on the sea-ice related aspects of this study, assuming that the underlying statistical method is correct. Doing so, I strongly second the major concern of the previous reviewer that the focus on a specific year of an ice-free Arctic Ocean is misleading. Indeed, I recommend leaving out this discussion altogether. There are several reasons for this assessment:

First, time is not a driver of the state evolution of the system. Hence, any reference to a specific year will always depend on the trajectory of future emissions which is highly uncertain. Focusing instead on the actual drivers of the system, be it temperature or CO₂ emissions, allows one to obtain a trajectory-independent answer as to the conditions under which the Arctic might become sea-ice free in summer.

Second, in contrast to the reply by the authors to the previous reviewer's comments, the real-world GMST at which the ice will effectively disappear is barely sensitivity to the trajectory of future emissions or future temperature evolution. This is among others shown by the very robust linear relationship between global-mean or Arctic temperature and Arctic sea-ice coverage that is robust in all CMIP5 models and the observations, and also by the linear relationship between emissions and Arctic sea-ice coverage, which again is robust in all CMIP5 models and the observational record independent of trajectory (see, for example, Notz and Stroeve, 2016, <https://doi.org/10.1126%2Fscience.aag2345>). In contrast, the sea-ice evolution is by no means a linear function of time as apparently implied here.

We thank the reviewer for their advice. Taking this into account we leave out the discussion on the Arctic ice-free year.

I hence question the validity of the analysis carried out here regarding the future evolution of sea ice as a function of time. It is obvious from physical principles that the two main control variables on the future evolution of the ice cover are (a) how much ice is there just now (which explains the correlation with SIE examined by the authors), and (b) how fast does the ice disappear (the "correctness" of which in individual models is apparently neglected here). As pointed out for example by Rosenblum and Eisenman (2017, <https://doi.org/10.1175%2FJCLI-D-16-0455.1>) or Notz and Stroeve (2016) almost all CMIP5 models have too low a sensitivity of past and future ice loss, so any study neglecting the impact of the sensitivity will most likely underestimate the speed of future sea-ice loss. Note, in addition, that Stroeve and Notz (2016, <https://doi.org/10.1016%2Fj.gloplacha.2015.10.011>) showed that reasonable past

performance is no indication for reasonable future performance.

Ignoring the details of the statistical analysis, I believe that the following two questions must robustly be answered before the method itself can be applied:

1. Do we capture all relevant parameters of "model quality" when weighting the "correctness" of any given model?
2. Are our assumption on the physical relationship between our control variable (here: time) and the driven parameter (here SIE) robust?

Both questions can probably be much easier by answered by "yes" if one examines sea-ice evolution as a function of either temperature or emissions, including a simultaneous assessment of the quality of current SIE and of the models' sensitivity to warming or to emissions.

I fear that the method might erroneously be judged as wrong if the paper were published in the present form, because we know with great certainty that the estimated PDFs of when the Arctic sea ice might be gone are erroneous. You will be hard pushed to find any sea-ice scientist who could even remotely imagine to have an Arctic sea-ice cover to exist beyond 2060 in an RCP8.5 scenario...

As it stands, I would trust results from carefully bias corrected model simulations (e.g., reference [31], or Sigmond et al., 2018, <https://doi.org/10.1038%2Fs41558-018-0124-y>), or from studies which exploit the linearity between forcing and sea ice (ref. 32], far more than I would trust a fairly different result from the approach taken here because I believe that the input data used here, and the underlying assumptions on physical relationships, are not robust. Hence, if the authors find very different ranges of future sea-ice evolution than other recent studies, these differences should be explained based on our physical understanding of the system.

We thank the reviewer for their pointers on this. We have re-done the analysis to incorporate (i) both mean SIE, and SIE sensitivity to temperature, as constraints on the models, and (ii) now use only GMST change to melt as the projection variable.

Other issues I noted:

There is somewhat misleading notion of a possible "model correctness" in the current version of the paper. However, we know that all our climate models are wrong - "and not even approximately true". (Parker, 2009, <https://doi.org/10.1111%2Fj.1467-8349.2009.00180.x>). They can nevertheless be very useful to answer specific questions, which is what we use them for. It would be helpful to have this line of thinking reflected in this paper.

We agree with the reviewer. We have inserted a small discussion on lines 112-117, and modified the text throughout the paper accordingly.

Recently, Olonscheck and Notz (2017, <https://doi.org/10.1175%2FJCLI-D-16-0428.1>) established a notion of "model consistency", which implies that the deviation of a specific model from an observational record is consistent as long as the deviation is smaller than the model's simulated internal variability (or standard deviation). They found a wide range in sea-ice variability across models, in contrast to the claim in line 204 (version with marked changes) that the standard deviation is constant across all models. Ideally, the models' different standard deviation should for the present study be estimated from the control simulation and then used for the statistical analysis for each model individually. Realistically, it should at least be noted that this should be done for a more robust assessment :-)

We understand the reviewer's thinking. Our approach for present-day mean SIE constraint already accounts for the different internal variability in the models. Olonscheck and Notz (2017) use a metric that compares sample mean model output and sample mean observed output, which are then divided by a measure of internal variability [1]. What we are doing, on the other hand, is comparing a "true" unobserved population mean of observations with the corresponding "true" mean of the models. These two quantities are unknown parameters, represented by pdfs, from which we draw samples. These quantities are distinct from sample means. All pdf samples with the difference between the population means within some tolerance correspond to model adequacy hypothesis being true. If, for a particular model, the internal variability (represented by the standard deviation) is high, this is expected to result in a broader pdf of the "true" mean for a particular model. This is expected to affect the posterior probability of that model.

However, the new constraint of SIE sensitivity does not account for the different internal variability. This is now discussed in the caveats section (lines 644-652).

Figure 3: If the colors mean nothing, I strongly suggest to not vary them.
Thank you. We have made all relevant colors gray.

I hope the authors will find these comments helpful in re-doing the analysis of future Arctic sea-ice evolution. The method suggested here seems powerful and, if robust, would be very welcome by the community. Using a misleading example to sell its usefulness could overshadow the beauty of the underlying statistical approach.

Best wishes.

Dirk Notz

We thank the reviewer for their positive comment on our statistical method. We hope the revision has improved the manuscript considerably, and that the reviewer will enjoy reading the resubmitted version.

References:

1. Olonscheck D, Notz D. Consistently Estimating Internal Climate Variability from Climate Model Simulations. J Clim. 2017 Sep 1;30(23):9555–73.

REVIEWERS' COMMENTS:

Reviewer #4 (Remarks to the Author):

Dear authors,

I am impressed by how much the application part of this paper has been improved in response to my previous comments. I find this a truly worthwhile and interesting study. Based on my review of the application part of this paper, I can recommend a publication with no further changes.

All the best,

Dirk Notz